# A Principle of Targeted Intervention for Multi-Agent Reinforcement Learning

**Anjie Liu**[*]
Thrust of Artificial Intelligence
HKUST (GZ)

**Jianhong Wang**[*][†]
INFORMED-AI Hub
University of Bristol

**Samuel Kaski**[‡]

ELLIS Institute Finland

**Jun Wang**
Centre for Artificial Intelligence
University College London

**Mengyue Yang**
School of Engineering Mathematics and Technology
University of Bristol

## Abstract

Steering cooperative multi-agent reinforcement learning (MARL) towards desired outcomes is challenging, particularly when the global guidance from a human on the whole multi-agent system is impractical in a large-scale MARL. On the other hand, designing external mechanisms (e.g., intrinsic rewards and human feedback) to coordinate agents mostly relies on empirical studies, lacking a easy-to-use research tool. In this work, we employ multi-agent influence diagrams (MAIDs) as a graphical framework to address the above issues. First, we introduce the concept of MARL interaction paradigms (orthogonal to MARL learning paradigms), using MAIDs to analyze and visualize both unguided self-organization and global guidance mechanisms in MARL. Then, we design a new MARL interaction paradigm, referred to as the targeted intervention paradigm that is applied to only a single targeted agent, so the problem of global guidance can be mitigated. In implementation, we introduce a causal inference technique—referred to as Pre-Strategy Intervention (PSI)—to realize the targeted intervention paradigm. Since MAIDs can be regarded as a special class of causal diagrams, a composite desired outcome that integrates the primary task goal and an additional desired outcome can be achieved by maximizing the corresponding causal effect through the PSI. Moreover, the bundled relevance graph analysis of MAIDs provides a tool to identify whether an MARL learning paradigm is workable under the design of an MARL interaction paradigm. In experiments, we demonstrate the effectiveness of our proposed targeted intervention, and verify the result of relevance graph analysis.

## 1 Introduction

Multi-agent reinforcement learning (MARL) provides a powerful framework for sequential decision-making in complex interacting systems, applied in autonomous systems including robotics [1, 2, 3]. A central challenge in MARL is enabling effective coordination among agents that must operate with partial information in dynamic environments. Achieving robust coordination is often hindered by inherent difficulties such as non-stationarity [4], where concurrent learning destabilizes individual agent's perspectives, making stable learning difficult. Furthermore, effective coordination often requires guiding a multi-agent system towards a desired outcome—an outcome with specific beneficial properties that standard MARL approaches may struggle to achieve consistently—especially when only maximizing a single, shared team reward (task goal) [5, 6].

---

[*]Equal contributions, names listed in alphabetical order. [†]Correspondence to Jianhong Wang (`jianhong.wang@bristol.ac.uk`). [‡]Samuel Kaski is also with Aalto University and University of Manchester.

To address these challenges in MARL, researchers often introduce external mechanisms that guide agent interaction and learning—what we refer to as MARL interaction paradigms—to enhance performance, efficiency, and applicability. Intrinsic rewards, for instance, can improve sample efficiency and direct agent behaviour by encouraging exploration or coordination [7, 8, 9]. Another approach incorporating human feedback, leverages human expertise to accelerate learning. This helps agents tackle objectives that are complex to specify via simple reward functions, and aligns agent behaviour to desired norms or human values [10, 11, 12].

*These MARL interaction paradigms go beyond the scope of the conventional MARL learning paradigms (e.g., centralized training and decentralized execution (CTDE) [13][Ch. 9.1.3]).*

While beneficial, applying global guidance to the entire multi-agent system in MARL is often impractical due to complexity, costs [14] and safety concerns [15]. Consider, for example, autonomous vehicles navigating a complex intersection or merging onto a highway. Ensuring safe and efficient passage requires intricate coordination. Also, providing simultaneous, specific feedback or instructions to all vehicles is infeasible, due to challenges such as complex safety validation and lack of universal communication protocols [16]. Yet, effective group coordination remains essential. An intriguing possibility arises if an additional desired outcome as a guidance is applied to a single targeted vehicle, e.g., instructing it to adjust its speed or yield at a critical moment. The resulting behaviour can serve as a pattern, enabling surrounding vehicles to coordinate. This scenario highlights a critical challenge where global guidance is essential but prohibitively difficult or costly, leading directly to the central question motivating our work:

*Can effective coordination be achieved when assigning an additional desired outcome to a single targeted agent, relying on its influence over the rest of the agents in a multi-agent system?*

Addressing the challenge of designing effective targeted intervention requires a principled framework to model and analyze how guiding a single agent impacts the overall multi-agent system. To this end, we propose leveraging multi-agent influence diagrams (MAIDs) [17], as a formal graphical language to encode complex strategic dependencies and information flow within the MARL learning process. Crucially, their bundled relevance graphs enable the analysis of a key property that we refer to as solvability. The identification of solvability guides better understanding and prediction of MARL interaction paradigms. To answer our research question, we utilize MAIDs to design the targeted intervention paradigm that intervenes on a single targeted agent, which reliably guides the entire multi-agent system towards a desired outcome. Formally, we frame this problem as selecting a preferred Nash equilibrium (NE) from many possible equilibria [18], where the preferred NE is precisely the one that ensures effective coordination with satisfying an additional desired outcome.

The main contributions of this paper are summarized as follows: (1) We introduce the concept of MARL interaction paradigms, using MAIDs to analyze and visualize both unguided self-organization and global guidance mechanisms in MARL. Then, we propose a new MARL interaction paradigm, referred to as the targeted intervention paradigm, where only a single targeted agent is intervened on. (2) The solvability of existing MARL learning paradigms under various MARL interaction paradigms can be analyzed, with the help of visualizing the corresponding relevance graphs. (3) Since MAIDs can be viewed as causal diagrams, we draw on the principle of causal inference to implement the targeted intervention paradigm, which we refer to as the Pre-Strategy Intervention (PSI). It is provable that the PSI can reach a composite desired outcome consisting of the primary task goal for coordination and an additional desired outcome. (4) The PSI is implemented as a pre-policy module that can be integrated into generic MARL algorithms.

The proposed PSI is evaluated in Multi-Agent Particle Environment (MPE) and Hanabi [19, 20]. Furthermore, the solvability of different MARL learning paradigms concluded from relevance graphs are verified in experiments. Our code is publicly available as an open-source repository.[2]

## 2 Multi-Agent Influence Diagrams (MAIDs)

We now review *multi-agent influence diagrams* (MAIDs) [17], which is an augmentation of the Bayesian network to describe multi-agent decision making to maximize their utility. In the rest of the paper, the terms *variable* and *node* will be used interchangeably. An MAID is usually described as a tuple $\mathcal{M} = (\mathcal{I}, \mathcal{X}, \mathcal{D}, \mathcal{U}, \mathcal{G}, Pr)$, where $\mathcal{I}$ is a set of agents. $\mathcal{X}$ is a set of *chance variables*

---

[2]https://github.com/iamlilAJ/Pre-Strategy-Intervention

indicating decisions of nature. Each chance variable $X \in \mathcal{X}$ is associated with a set of parents $Pa(X) \subset \mathcal{X} \cup \mathcal{D}$. The $\mathcal{D} := \bigcup_{i \in \mathcal{I}} \mathcal{D}_i$ is a set of all agents' decision variables, where $\mathcal{D}_i$ is the set of agent $i$'s decision variables. For a decision variable $D \in \mathcal{D}_i$, $Pa(D)$ is the set of variables whose values are informed to agent $i$ when it selects a value of $D$. $\mathcal{U} := \bigcup_{i \in \mathcal{I}} \mathcal{U}_i$ is a set of utility variables, where $\mathcal{U}_i$ is agent $i$'s utility variable as its utility function. Note that utility variables cannot be parents of other variables. A directed acyclic graph $\mathcal{G}$ with variables $\mathcal{V} = \mathcal{X} \cup \mathcal{D} \cup \mathcal{U}$ is formed. $Pr$ is a conditional probability distribution (CPD) defined over chance variables $X$ such as $Pr(X|Pa(X))$, and utility variables $U \in \mathcal{U}$ such as $Pr(U|\mathbf{pa})$, for each $\mathbf{pa} \in dom(Pa(U))$. Note that $Pr(U|Pa(U))$ is a Dirac function (i.e. $U$ is a deterministic function). In other words, for each instantiation $\mathbf{pa} \in dom(Pa(U))$, there is a value of $U$ that is assigned probability 1, and probability of other values is 0. To simplify the notation, $U(\mathbf{pa})$ denotes as the value of $U$ that has probability 1 when $Pa(U) = \mathbf{pa}$. The total utility that an agent $i$ obtained from an instantiation of $\mathcal{V}$ is the sum of the values of $\mathcal{U}_i$, i.e. $\sum_{U \in \mathcal{U}_i} U(\mathbf{pa})$ where $\mathbf{pa} \in dom(Pa(U))$.

**Decision Rule and Strategy.** An agent makes a decision at variable $D$ depending on its $Pa(D)$, determined by a *decision rule* $\delta : dom(Pa(D)) \to \Delta(dom(D))$. $\Delta$ indicates the probability distribution space over a set. An assignment $\sigma$ of decision rules to each decision $D \in \mathcal{D}$ is called a *strategy profile*. A partial strategy profile $\sigma_{\mathcal{E}}$ is an assignment of decision rules to a subset of $\mathcal{D}$, as a restriction of $\sigma$ to $\mathcal{E}$, and $\sigma_{-\mathcal{E}}$ denotes the restriction of $\sigma$ to variables in $\mathcal{D} \backslash \mathcal{E}$. The assignment of $\sigma_{\mathcal{E}}$ to the MAID $\mathcal{M}$ induces a new MAID denoted by $\mathcal{M}[\sigma]$, and each $D \in \mathcal{E}$ would become a chance variable with the CPD $\sigma(D)$. When $\sigma$ is assigned to every decision variable in MAID, the induced MAID becomes a Bayesian network with no more decision variables. This Bayesian network defines a joint probability distribution $P_{\mathcal{M}[\sigma]}$ over all the variables in $\mathcal{M}$.

**Expected Utility and Nash Equilibrium.** Given a strategy profile assigned to each decision variable, with the resulting joint probability distribution $P_{\mathcal{M}[\sigma]}$ and suppose that $\mathcal{U}_i = \{U_1, ..., U_m\}$, we can write the expected utility for an agent $i$ as

$$\mathbb{E}U_i(\sigma) = \sum_{(u_1,...,u_m) \in dom(\mathcal{U}_i)} P_{\mathcal{M}[\sigma]}(u_1, ..., u_m) \sum_{k=1}^{m} u_k. \tag{1}$$

Given Eq. (1), the strategy $\sigma_{\mathcal{E}}^*$ is optimal for $\sigma$, for a subset $\mathcal{E} \subset \mathcal{D}_i$, if $\mathbb{E}U_i((\sigma_{-\mathcal{E}}, \sigma_{\mathcal{E}}^*)) \geq \mathbb{E}U_i((\sigma_{-\mathcal{E}}, \sigma_{\mathcal{E}}'))$, as shown in Definition 2.1. Furthermore, if for all agents $i \in \mathcal{I}$, $\sigma_{\mathcal{D}_i}$ is optimal for the strategy profile $\sigma$, then $\sigma$ is a Nash equilibrium (NE), as shown in Definition 2.2.

**Definition 2.1 (Optimal Strategy** [17]**).** Let $\mathcal{E}$ be a subset of $\mathcal{D}_i$, and let $\sigma$ be a strategy profile. $\sigma_{\mathcal{E}}^*$ is optimal for the strategy profile $\sigma$ if, in the induced MAID $\mathcal{M}[\sigma_{-\mathcal{E}}]$, where the only remaining decisions are those in $\mathcal{E}$, the strategy $\sigma_{\mathcal{E}}^*$ is optimal, for all strategies $\sigma_{\mathcal{E}}'$, such that $\mathbb{E}U_i((\sigma_{-\mathcal{E}}, \sigma_{\mathcal{E}}^*)) \geq \mathbb{E}U_i((\sigma_{-\mathcal{E}}, \sigma_{\mathcal{E}}'))$.

**Definition 2.2 (Nash Equilibrium** [17]**).** A strategy profile $\sigma$ is a Nash equilibrium for a MAID $\mathcal{M}$ if for all agents $i \in \mathcal{I}$, $\sigma_{\mathcal{D}_i}$ is optimal for the strategy profile $\sigma$.

*Remark* 2.3. For each MAID there can be multiple NEs (corresponding to multiple strategy profiles). We denote the random variable describing a possible NE over a set of NEs, $\{\hat{\sigma}_1, \ldots, \hat{\sigma}_k\}$ as $\hat{\boldsymbol{\sigma}}$. For any $\hat{\sigma} \in dom(\hat{\boldsymbol{\sigma}})$, we define the probability for an arbitrary NE as $P_\sigma(\hat{\sigma}) := Pr(\hat{\sigma}_{D_1}, \ldots, \hat{\sigma}_{D_i}, \ldots, \hat{\sigma}_{D_n})$, where $n := |\mathcal{I}|$ is the number of agents in the MAID. The probability of a strategy profile is defined as the joint probability that each agent $i$ plays some strategy on the agent's decision variable $D_i$.

**Definition 2.4 (S-Reachability** [17]**).** A node $D'$ is s-reachable from a node $D$ in a MAID $\mathcal{M}$ if there is some utility node $U \in \mathcal{U}_D$ such that if a new parent $\hat{D}'$ were added to $D'$, there would be an active path (Example C.1 and Definition C.1 in Appendix C.2) in $\mathcal{M}$ from $\hat{D}'$ to $U$ given $Pa(D) \cup \{D\}$, where a path is active in a MAID if it is active in the same graph, viewed as a Bayesian network (i.e. it is not blocked as per the standard d-separation rules). The relevance graph for a MAID $\mathcal{M}$ is a directed graph whose nodes are the decision nodes of $\mathcal{M}$, and which contains an edge $D' \to D$ if and only if $D'$ is s-reachable from $D$.

**Relevance Graph.** A *relevance graph* as shown in Definition 2.4 defines a directed graph induced from an MAID, describing the binary relation between two decision variables. If there exists an edge $D' \to D$, it implies that the decision variable $D$ is *strategically relies* on another decision variable $D'$. In other words, the decision rules for $D'$ is required to evaluate the decision rules for $D$. If there exist both $D' \to D$ and $D \to D'$, then the relevance graph is cyclic. Furthermore, if $D$ and $D'$

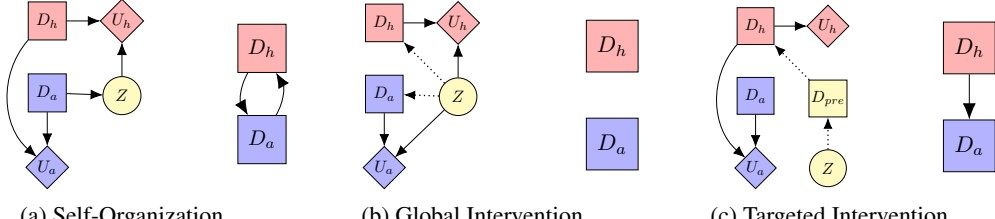

(a) Self-Organization.  (b) Global Intervention.  (c) Targeted Intervention.

Figure 1: MAIDs (left part in each figure) and relevance graphs (right part) for one-shot simultaneous-move MARL interaction paradigms: (a) **self-organization**, where agents coordinate with no external mechanism; (b) **global intervention**, where a coordinator provides signals influencing all agents and their utilities; (c) **targeted intervention**, where a pre-strategy intervention applied on a targeted agent. In these diagrams, squares represent decision variables (specifically, red for the targeted agent, blue for other agent, yellow for a pre-decision); diamonds represent utility variables; and circles with $Z$ describe signals to guide agents for achieving their desired outcomes represented as utility variables. Relevance graphs depict strategic dependencies between these decision variables.

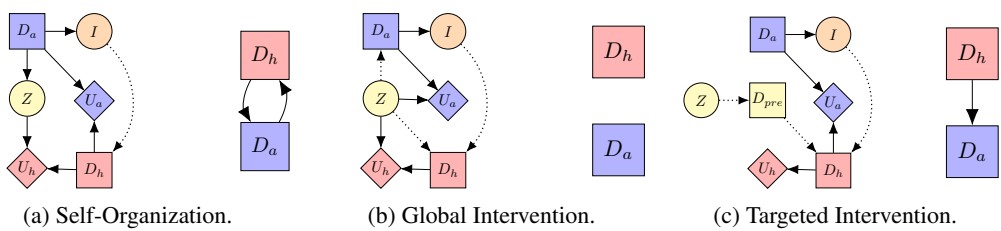

(a) Self-Organization.  (b) Global Intervention.  (c) Targeted Intervention.

Figure 2: MAIDs and their relevance graphs for one-shot sequential-move games. In addition to the variables introduced in Figure 1, the orange circles with $I$ indicate the information sets.

belong to two agents respectively, their payoffs depend on the decisions at both $D$ and $D'$. In this case, the optimality of one agent's decision rule is intertwined with another agent's decision rule [17]. Therefore, agents seeking decision rules **individually** to reach an NE, analogous to independent learning [13][Ch. 9.3], is unsolvable, because mismatch between agents' decision rules could happen. *Remark* 2.5. One way to make cyclic relevance graphs solvable is making agents' decision rules matched via seeking agents' decision rules **collectively** [17], analogous to CTDE [13][Ch. 9.1.3].

## 3 Multi-Agent Reinforcement Learning Interaction Paradigms

We introduce the concept of MARL interaction paradigms (orthogonal to MARL learning paradigms) to describe the ways in which agents' interactions are structured—ranging from unguided self-organization to externally guided mechanisms. Specifically, we first formalize two existing MARL interaction paradigms—self-organization and global intervention [21, 22, 23]—using multi-agent influence diagrams (MAIDs), and then introduce our proposed targeted intervention paradigm.

### 3.1 Formalizing MARL Interaction Paradigms

By the definition of MAIDs in Section 2, we model MARL interaction paradigms based on agent variables ($\mathcal{I}$), decision variables ($\mathcal{D}$), chance variables ($\mathcal{X}$) representing states and information, and utility variables ($\mathcal{U}$) representing objectives. Utility variables capture the desired outcomes that agents pursue, whereas the special chance variables $\mathcal{Z} \subset \mathcal{X}$ describe signals to guide agents for achieving their desired outcomes. The formal definition is delineated in Definition 3.1.

**Definition 3.1.** An MARL interaction paradigm can be specified as an MAID $(\mathcal{I}, \mathcal{X}, \mathcal{D}, \mathcal{U})$. $\mathcal{I}$ is the set of agents. $\mathcal{X}$ is the set of chance variables. $\mathcal{D} = \bigcup_{i \in \mathcal{I}} \mathcal{D}_i$ is the set of agent decision variables. $\mathcal{U}$ are utility variables representing objectives. The set of special chance variables ($\mathcal{Z} \subset \mathcal{X}$) describe signals to guide agents for achieving their desired outcomes.

We now define three MARL interaction paradigms as follows:

(1) **Self-Organization.** This paradigm refers to the MARL setting where agents coordinate solely via their direct observations of the environment, without any external mechanism to steer them towards their desired outcomes. Each agent must independently elicit a signal (a pattern of behaviours) based on local information to influence the other agents towards its desired outcome. For clarity of illustration, we focus on analyzing guidance to a single agent. We visualize this implicit process as a path from decision variables $D_a$ to the signal $Z$, and then to the utility nodes $U_h$, illustrated by MAID structures in Figure 1a and 2a.

(2) **Global Intervention.** This paradigm features an external, centralized coordination providing simultaneous explicit guidance signals to all agents. In the MAID, as shown in Figure 1b and 2b, this is represented as information links from $Z$ to agents' decision variables, and $Z$ usually influence their utility nodes (e.g., $U_a$ and $U_h$) directly, shaping their desired outcomes.

(3) **Targeted Intervention.** This paradigm involves applying an external guidance signal to a targeted agent, as shown in Figure 1c and 2c. For conciseness, we illustrate this as one of its implementation, referred to as pre-strategy intervention (Section 3.3), where $D_{pre}$ receives a guidance signal $Z$ and outputs filtered information. This intervention modifies the behaviours of the targeted agent, which in turn indirectly influences other agents. As a result, agents coordinate to achieve their desired outcomes provided by the guidance signal $Z$. While the concept of targeted intervention paradigm could potentially apply to multiple targeted agents with multiple desired outcomes, this paper specifically focuses on the case of a single targeted agent.

## 3.2   Interpreting MARL Interaction Paradigms

We now demonstrate how MAIDs interpret the three MARL interaction paradigms we mentioned above. Each MAID elucidates the structure of a paradigm (through the role of variable $Z$) and its influence on decisions and utilities. Subsequently, we analyze how these structural differences translate into various strategic dependency patterns, as revealed by their relevance graphs.

Analyzing the relevance graph (Definition 2.4) corresponding to each MAID, reveals the key solvability property[3] of MARL algorithms. As shown in Figure 1a and 2a, the relevance graphs for the self-organization paradigm are cyclic, implying computational or theoretical intricacies in finding solutions. In contrast, the relevance graphs for both the global intervention paradigm (Figures 1b and 2b) and targeted intervention paradigm (Figures 1c and 2c) are acyclic, suggesting they may be solvable by a broader class of MARL algorithms. It is possible in global intervention that agents learn independently based solely on the external central signal, in which case the relevance graph would have no edges at all, indicating no policy dependency [24]. While both global intervention and targeted intervention paradigms offer acyclic structures, our proposed method relying on the targeted intervention paradigm provides a distinct advantage: it can facilitate solvability and effectiveness only by intervention on a single targeted agent, offering a more practical approach in contrast to the global intervention paradigm.

## 3.3   Pre-Strategy Intervention: An Implementation of Targeted Intervention Paradigm

This section introduces the pre-strategy intervention—a principled design of the targeted intervention paradigm. By applying a pre-strategy intervention, we aim to guide the multi-agent system to a preferred Nash equilibrium (NE), selecting one with an additional desired outcome from the multiple equilibria that satisfy the primary task goal. Our approach first defines the pre-strategy intervention, then details how its influence on reaching the preferred NE is quantified and optimized via learning a pre-policy to generate pre-strategies intervening on a single, targeted agent.

### 3.3.1   Definition of Pre-Strategy Intervention

To regulate the intervened agent to maximize the desired outcome $U$ that captures the preferred NE, we propose to add a *pre-decision variable* $D_{pre}$ as a new parent to the decision variable $D_h$ of a selected agent $h \in \mathcal{I}$. In analogy to decision variables in MAIDs, we need to assign a strategy $\sigma_{pre}$ which we refer to as *pre-strategy*, and this operation is named *pre-strategy intervention*. This definition respects the convention of *stochastic intervention* in *causal Bayesian networks* [25][Ch. 4].

---

[3] Seeking equilibria according to processing decision variables in s-reachability order (influencers first) [17].

A pre-strategy is determined by a *pre-policy* denoted by $\delta_{pre}$, which processes the agent's information and the guidance signal, as detailed in Definition 3.2.

**Definition 3.2** (**Pre-Strategy Intervention**). For a decision variable $D \in \mathcal{D}$ in a MAID, a pre-strategy intervention is an operation assigning a pre-strategy $\sigma_{pre}$ to a new parent $D_{pre}$ added to $D$, referred to as a pre-decision variable. The pre-strategy $\sigma_{pre}$ is determined by a pre-policy $\delta_{pre} : dom(Pa(D)) \times \mathcal{Z} \to \Delta(dom(\sigma_{pre}))$, where $\mathcal{Z}$ is the space of guidance signals representing an additional desired outcome.

### 3.3.2 Causal Effect of Pre-Strategy Intervention

The preferred NE is identified by maximizing a composite desired outcome, formally defined by a total utility variable $U_{tot} := U_{task} + U_{sec}$, comprising the utility $U_{task}$ of the primary task goal that all agents coordinate to achieve and the utility $U_{sec}$ of an additional desired outcome (a secondary goal) only assigned to a single, targeted agent.[4] This preferred NE is one of multiple equilibria that all yield the identical, maximum value of the primary task utility $U_{task}$. We formalize the causal effect of a pre-strategy intervention in Definition 3.3. The causal effect quantifies the total probabilities of $U_{tot} = u^*$, where $u^*$ is the utility value of the desired composite outcome, under a pre-strategy intervention for the additional desired outcome. In many cases, it is difficult to find a pre-strategy intervention that induces the preferred NE. Instead, we seek a pre-strategy that induces a probability distribution over a reduced set of NEs covering the preferred NE, denoted as $\hat{\boldsymbol{\sigma}}_{\mathcal{I}}$. Proposition 3.4 proves pre-strategy intervention maximizing this causal effect is guaranteed to exist.

**Definition 3.3** (**Causal Effect of Pre-Strategy Intervention**). A pre-strategy intervention is applied to a targeted agent's strategy profiles, which influences the total utility $U_{tot} = u^*$ associated with a Nash equilibrium (NE) $\hat{\sigma}$ after the pre-strategy intervention. The set of all possible NEs is denoted by $\hat{\boldsymbol{\sigma}}$ (with no intervention). Given that $\hat{\boldsymbol{\sigma}}_{\mathcal{I}}$ denotes a reduced set of NEs induced by a pre-strategy intervention, covering the preferred NE (aligned to the additional desired outcome), the causal effect measuring this pre-strategy intervention is defined as follows:

$$\Delta_{\text{CE}}^{\sigma_{pre}}(U_{tot} = u^*) = \underbrace{\int_{\hat{\sigma} \in \hat{\boldsymbol{\sigma}}_{\mathcal{I}}} P_{\mathcal{M}[\hat{\sigma}]}(U_{tot} = u^*) P_{\hat{\sigma}}(\hat{\sigma}) \, d\hat{\sigma}}_{P_{\mathcal{I}}(U_{tot}=u^*)} - \underbrace{\int_{\hat{\sigma} \in \hat{\boldsymbol{\sigma}}} P_{\mathcal{M}[\hat{\sigma}]}(U_{tot} = u^*) P_{\hat{\sigma}}(\hat{\sigma}) \, d\hat{\sigma}}_{P_{\mathcal{U}}(U_{tot}=u^*)} . \tag{2}$$

In Eq. (2), $P_{\mathcal{M}[\hat{\sigma}]}(U_{tot} = u^*)$ represents the likelihood of the total utility $U_{tot} = u^*$ under an arbitrary NE $\hat{\sigma}$. $P_{\hat{\sigma}}(\hat{\sigma})$ denotes the probability measure on an arbitrary NE $\hat{\sigma}$.

**Proposition 3.4.** *Given a MAID $\mathcal{M}$, assume that the function $P_{\mathcal{I}}$ in Eq. (2), representing the probability of observing $U_{tot} = u^*$ under a pre-strategy intervention, is upper semicontinuous and defined on a compact domain $\text{dom}(\sigma_{pre}) \subseteq \mathbb{R}^m$. There exists at least a pre-strategy intervention on an agent that does not decrease the probability of $U_{tot} = u^*$. Furthermore, there exists a pre-strategy intervention that maximizes the causal effect.*

To practically evaluate possible strategy profiles of the pre-strategies generated by a pre-policy, we reshape $P_{\mathcal{I}}$ as the following expression:

$$P(U_{tot} = u^* \mid \text{do}(\sigma_{pre})) = \sum_{\hat{\sigma} \in \hat{\boldsymbol{\sigma}}} P_{\mathcal{M}[\hat{\sigma}]}(U_{tot} = u^*) P_{\hat{\sigma}}(\hat{\sigma} \mid \text{do}(\sigma_{pre})). \tag{3}$$

Note that $P_{\mathcal{U}}$ is a constant with respect to the intervention, so it can be ignored during optimization. In Eq. (3), $P_{\hat{\sigma}}(\hat{\sigma} \mid \text{do}(\sigma_{pre}))$ is a probability distribution whose support includes *all possible strategy profiles*, which is feasible to implement and induce any possible $\hat{\boldsymbol{\sigma}}_{\mathcal{I}}$.

## 4 MARL for Sequential Decision Making in MAIDs

We now show how the pre-strategy intervention introduced above is implemented in MARL for sequential decision making. We model MARL using team reward Markov games [26], and the team reward is represented as the team utility variable. The team utility variable is defined as the

---

[4]The additional desired outcome is constrained to be non-conflicting with the primary task goal.

total utility variable to identify the preferred Nash equilibrium. The main idea is establishing a connection that a team reward Markov game can be transformed into a multi-agent influence diagram (MAID), allowing our framework to apply. The formal definition of a team reward Markov game and the detailed process of representing it as an MAID by matching their respective variables are provided in Appendix G. Building on this foundation, the subsequent subsections will extend the MARL interaction paradigms introduced in the previous section to team reward Markov games (Section 4.1), discuss the solvability of MARL learning paradigms under these MARL interaction paradigms (Section 4.2), and specify how the pre-strategy intervention in Eq. (3) is integrated into generic MARL algorithms (Section 4.3).

## 4.1 Extending MARL Interaction Paradigms to Team Reward Markov Games

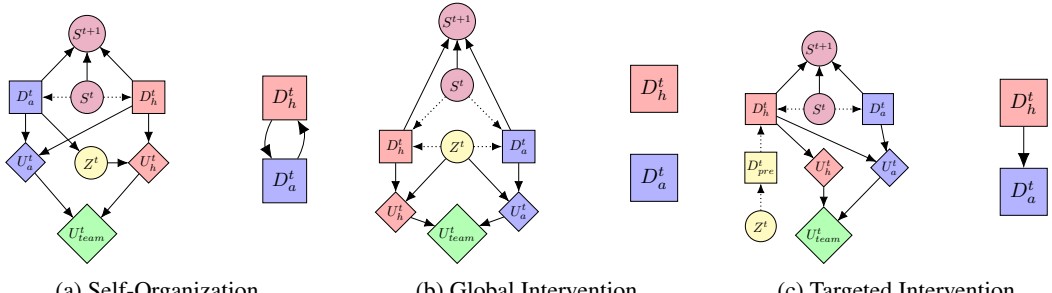

(a) Self-Organization.      (b) Global Intervention.      (c) Targeted Intervention.

Figure 3: MAIDs and their relevance graphs for dynamic simultaneous-move games. For conciseness, we only demonstrate one transition from timesteps $t$ to $t + 1$. The variables used in one-shot decision making (see Figure 1) are decorated with timesteps. The semantics of these variables remain unchanged. The pink circles with $s_t$ and $s_{t+1}$ indicate states for timesteps $t$ and $t + 1$. The green diamonds with $U_{team}^t$ indicate the team utility variable that represents the team reward. The incoming edge from an individual utility variable ($U_h^t$ and $U_a^t$) to the team utility variable indicates that an agent's individual contribution to the team reward.

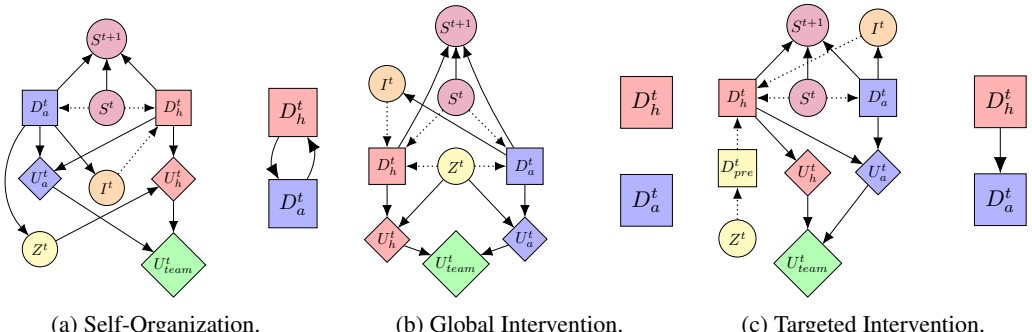

(a) Self-Organization.      (b) Global Intervention.      (c) Targeted Intervention.

Figure 4: MAIDs and their relevance graphs for dynamic sequential-move games. For conciseness, we only demonstrate one transition from timesteps $t$ to $t + 1$. In addition to the variables introduced in Figure 3, the orange circles with $I^t$ indicate the information sets.

We now demonstrate how to extend the three MARL interaction paradigms (Definition 3.1) to Markov games through the lens of MAIDs. Each interaction paradigm under sequential decision making in dynamic environments with simultaneous or sequential moves can be extended from one-shot decision making introduced in Section 3.2, by considering timesteps and states. Representing the team reward as the team utility variable $U_{team}^t$ that is an aggregation of agents' individual utility variables, we can derive the MAID graphical representation of these three interaction paradigms in team reward Markov games, which are illustrated in Figures 3 and 4.

It can be observed that the widespread global intervention paradigm implemented for MARL algorithms (e.g., applying a centralized coordinator [27, 28, 29]) can be justifiably interpreted with a visualization. More importantly, it opens a new door to design novel MARL interaction paradigms to solve a broad class of problems with the help of MAIDs, such as the targeted intervention paradigm proposed in this paper to solve the team reward Markov game with an additional desired outcome.

## 4.2 Solvability of MARL Learning Paradigms under MARL Interaction Paradigms

**Independent Learning.** Independent learning (IL) [13][Ch. 9.3] directly applies single-agent reinforcement learning algorithms to MARL. If the relevance graph is cyclic, individually optimizing each agent's decision variables using the generalized backward induction algorithm [17] cannot guarantee reaching a Nash equilibrium, even when the environmental model (MAIDs) is known—analogous to independent learning in the model-free MARL setting. This aligns with the **self-organization** paradigm (Figure 3a and 4a), reflecting the *non-stationarity dilemma* [1]. The **global intervention** (Figures 3b and 4b) and **targeted intervention** (Figures 3c and 4c) paradigms can generate *acyclic relevance graphs* that help mitigate non-stationarity, suggesting that augmenting IL with these two paradigms leads to better performance than its vanilla counterpart, becoming more solvable.[5]

**Centralized Training and Decentralized Execution.** As discussed in Remark 2.5, one way to address *cyclic relevance graphs* is enabling two agents' decision making matched. One solution is transforming a cyclic relevance graph to a *component graph*, where each *maximal strongly connected component* (SCC) is regarded as a supernode [30]. More specifically, a maximal SCC consists of all the decision variables that form a cyclic relevance graph at each timestep. Koller and Milch [17] showed that solving the acyclic component graph[6] using the generalized backward induction algorithm can reach a Nash equilibrium when the MAID model is provided. This is associated with the *centralized training and decentralized execution* (CTDE) [13][Ch. 9.1.3] applied in the model-free MARL setting. In summary, CTDE can be seen as an MARL learning paradigm to address the non-stationarity dilemma of vanilla IL under the **self-organization** paradigm, rather than switching to other MARL interaction paradigms.

## 4.3 Pre-Policy Learning in MARL

As we have established the MARL interaction paradigms for team-reward Markov games in MAIDs, the techniques developed in Section 3.3.2 can be applied to realize the pre-strategy intervention by defining the team utility variable $U_{team}^t$ as the total utility variable $U_{tot}^t$.[7] Following Definition 3.2, it requires learning a pre-policy $\delta_{pre} : dom(Pa(D)) \times \mathcal{Z} \to \Delta(dom(\sigma_{pre}))$ to generate the pre-strategy $\sigma_{pre}$ for the possible preferred Nash equilibria. This can be realized by maximizing the causal effect expressed in Eq. (3), with replacing the $U_{tot}$ by the cumulative team utilities over $L$ timesteps $\sum_{t=1}^{L} U_{tot}^t$, as per Eq. (1). The resulting algorithm, which can be integrated with generic MARL algorithms, is referred to as **Pre-Strategy Intervention** (PSI).

# 5 Experiments

## 5.1 Experimental Setups

We evaluate our proposed **Pre-Strategy Intervention** (PSI) in two environments: the Multi-Agent Particle Environment (MPE) [19], a simultaneous-move cooperative navigation game, and the Hanabi card game [20], which challenges coordination within a multi-agent system under partial observability. Hanabi is known for having multiple, distinct equilibria, making it an ideal testbed for analyzing convergence to a specific Nash equilibrium.[8] For all scenarios, we select a fixed agent to intervene on, which allows for a controlled analysis of the PSI. Full details of these environments, including specific experimental configurations and how additional desired outcomes are defined and implemented, are provided in Appendix H. In all environments, intrinsic returns are used for measuring reachability of

---

[5]The targeted intervention paradigm can, in principle, be addressed by a mixed learning paradigm with asynchronous updates, guided by the dependency of decision variables in the relevance graph: the targeted agent independently updates its policy, while the other agents update theirs under centralized training in the subsequent turn. Since such an approach is not yet a standard MARL learning paradigm, we leave it for future work.

[6]A component graph is always acyclic [31].

[7]By [32], maximizing each agent's equal contribution to a shared reward is equivalent to maximizing the reward. In the targeted intervention, the team reward is defined as $U_{tot}^t = U_{task}^t + U_{sec}^t$, where $U_{task}^t$ is shared among agents and $U_{sec}^t$ is only attributed to the targeted agent. Subsequently, the targeted agent's individual utility variable is defined as $U_{task}^t + U_{sec}^t$, while other agents' are defined as $U_{task}^t$. Thus, maximizing each agent's individual utility leads to the preferred NE among multiple equilibria of the primary task utility.

[8]We choose a convention called "5 Save" for the results in the main paper, while the results of another convention called "The Chop" is reported in Appendix I.2.2.

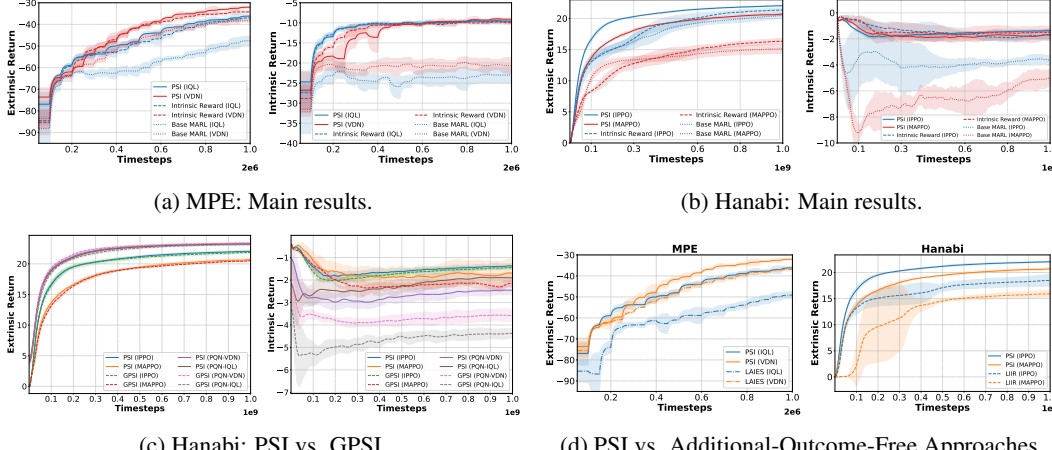

(a) MPE: Main results.

(b) Hanabi: Main results.

(c) Hanabi: PSI vs. GPSI.

(d) PSI vs. Additional-Outcome-Free Approaches.

Figure 5: In (a) and (b), PSI is compared against Intrinsic Reward and Base MARL; these plots show both extrinsic (the primary task completion) and intrinsic (reachability of an additional desired outcome) returns. In (c), the targeted intervention paradigm (PSI) is compared against the global intervention paradigm (GPSI). In (d), PSI is compared against approaches that only focus on the primary task completion with no consideration of any additional desired outcome.

additional desired outcomes, while extrinsic returns are used for measuring primary task completion. Results from 5 random seeds are reported as means with $95\%$ confidence intervals.

**Implementation of Our Method.** Our PSI is implemented by designing a pre-policy module (a GRU or MLP, matching agent's backbones), which takes a concatenated input of environmental observations and a measure for an additional desired outcome (formulated as an intrinsic reward).[9] It outputs an embedding, which is then fed to Q-value functions for value-based MARL methods or critics for policy-based methods.

**Baselines.** All baselines are implemented with the same architecture and training setups as our method. We evaluate our method against several base MARL algorithms referred to as Base MARL (self-organization paradigms): for MPE, these include IQL [33], VDN [34] and QMIX [35]; for Hanabi, we use IPPO [36], MAPPO [37] and PQN [38] (with PQN-IQL for IL and PQN-VDN for CTDE variants). An ablation study that removes the pre-policy module while retaining the intrinsic reward maximization, referred to as Intrinsic Reward, examines the effectiveness of the pre-policy module. To emphasize the strength of our targeted intervention paradigm, we also compare PSI against the global intervention version of PSI, referred to as GPSI, which applies the pre-policy module to intervene on **all** agents. Finally, we include baselines of the global intervention paradigm that **only** focuses on primary task completion, such as the modified versions of LIIR [8] and LAIES [29], as representative policy-based and value-based methods, respectively.

## 5.2 Results and Analysis

**Coordination Achieving When Intervening on a Single, Targeted Agent.** As Figure 5d shows, our PSI can outperform both LIIR and LAIES on the primary task completion in both MPE and Hanabi. This result confirms that coordination can be achieved when assigning an additional desired outcome (e.g. convention) derived from human knowledge, to a single targeted agent through the targeted intervention paradigm, answering our motivating research question. Moreover, this may even improve primary task completion, compared with approaches of the global intervention paradigm that does not consider any additional desired outcome, avoiding the issue of miscoordination.

**Verification of Relevance Graph Analysis.** Our MAID and relevance graph analysis predicts that PSI improves the solvability of MARL learning paradigms (Section 4.2), enabling the vanilla IL learning paradigm more solvable. Experiments verify this across different settings. In simultaneous-move MPE (analysis in Figure 3c and results in Figure 5a), IQL equipped with our PSI achieves task completion comparable to VDN, a CTDE algorithm. This is a notable improvement for IQL

---

[9]This implementation maintains solvability of MARL algorithms with the PSI (Appendix F).

over its baseline performance (without our PSI). Similarly, in sequential-move Hanabi (analysis in Figure 4c and results in Figure 5b), IL algorithms augmented with the PSI also achieve performance comparable to, or even better than CTDE algorithms.

**Targeted Intervention vs. Global Intervention.** Our targeted intervention paradigm (PSI) consistently outperforms the global intervention paradigm (GPSI), as shown in Figures 5c. Although both paradigms achieve comparable performance on the primary task completion, the global intervention paradigm often struggles to reach the additional desired outcome. We attribute this to the inherent practical difficulty of designing (learning) an effective coordination mechanism to globally assign beneficial, non-conflicting goals to multiple agents simultaneously. This highlights the potential application of PSI to safety-critical systems [39], which require strictly controlled operation.

**Ablation Study on the Pre-Policy Module.** To assess our pre-policy module's contribution, we conduct an ablation study to compare PSI against the Intrinsic Reward baseline. Experimental results presented for MPE in Figure 5a and for Hanabi in Figure 5b reveal a key distinction: while both PSI and the Intrinsic Reward baseline effectively lead to the attainment of the additional desired outcome, the PSI (with the pre-policy module) achieves notably superior performance on the primary task completion. This highlights that the pre-policy module is instrumental for retaining the targeted agent's adherence to an additional desired outcome while not forgetting the primary task goal.

**Analysis of Nash Equilibrium Convergence in Hanabi.** The Hanabi environment, with its multiple distinct equilibria [20], serves as an ideal testbed for analyzing convergence to a preferred Nash equilibrium (NE). We treat a specific human convention as a preferred, high-performing NE and design the intrinsic reward to directly measure the agents' compliance to it. As shown in Figure 5b, the high and stable intrinsic return achieved by our PSI provides a strong evidence that the agents successfully converge to the preferred NE. In contrast, the low intrinsic return of the baselines suggest they fail to establish the convention and are likely stuck in one of Hanabi's many inferior equilibria.

**Additional Results.** Appendix I details further exploration of our method's capabilities and robustness. These include: (1) evaluation using varied additional desired outcomes in MPE and Hanabi; and (2) performance assessments under noisy observation conditions.

## 6   Conclusion

**Summary.** We incorporate multi-agent influence diagrams (MAIDs) as a principled framework into designing and analyzing the targeted intervention paradigm in MARL. Our Pre-Strategy Intervention (PSI) approach as an implementation of the targeted intervention paradigm, applied to a single targeted agent, maximizes the causal effect on reaching a composite desired outcome that integrates both the primary task goal and an additional desired outcome. Furthermore, the relevance graphs of MAIDs offer theoretical insights into the solvability of MARL learning paradigms under various MARL interaction paradigms. Experiments verify the effectiveness of the PSI.

**Limitation.** Our principle currently presumes that the underlying structure of MARL interaction paradigms in MAIDs is complete or can be precisely modelled. This structural knowledge is a prerequisite for effective influence propagation via targeted intervention and can be challenging to define a priori in realistic environments or complex systems. Additionally, our analysis primarily focuses on a single targeted intervention, which may limit its applicability to real-world settings.

**Future Work.** Key future directions of our work include but are not limited to:

1. Learning MAID structures from data (e.g., via causal discovery [40]) to reduce reliance on pre-specified models. Further, the MAID can be seen as a foundation to describe the world model.
2. Coordinating a multi-agent system under the targeted intervention paradigm on multiple agents.
3. Designing the optimal criteria for selecting the number of targeted agents and appointing proper agents with various types to complete different tasks. The information-theoretic measure of empowerment [7, 41] offers a potential quantitative basis for such criteria.
4. Integrating advanced reasoning modules (e.g. large language models and the broader generative AI) to enhance the capability of the PSI on the targeted agent in online learning and online adaptation to unknown agent teammates and unseen situations [42].
5. Designing MARL algorithms with asynchronous updates and mixed learning paradigms inspired by the targeted intervention paradigm (Footnote 5).

## Acknowledgment

Jianhong Wang is supported by the Engineering and Physical Sciences Research Council (EPSRC) [Grant Ref: EP/Y028732/1]. Samuel Kaski is supported by UKRI Turing AI World-Leading Researcher Fellowship, EP/W002973/1.

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

# A   Notation

Table 1: Summary of key notations used in this paper.

| Notation | Description |
|---|---|
| **MAID Core Components** | |
| $\mathcal{M}$ | Multi-Agent Influence Diagram (MAID). |
| $\mathcal{I}$ | Set of agents in the MAID. |
| $\mathcal{X}$ | Set of chance variables (representing states, information, etc.). |
| $\mathcal{D}, \mathcal{D}_i$ | Set of decision variables for all agents, and for agent $i$. |
| $\mathcal{U}, \mathcal{U}_i$ | Set of utility variables (objectives) for all agents, and for agent $i$. |
| $\text{Pa}(D)$ | Parent set of a decision variable $D$. |
| $\mathcal{G}$ | Directed acyclic graph (DAG) of the MAID. |
| $Pr$ | Probability distribution. |
| $\sigma$ | A strategy profile (assignment of decision rules to all agents). |
| $\hat{\sigma}$ | An arbitrary Nash equilibrium (NE). |
| $\boldsymbol{\hat{\sigma}}$ | The set of all possible NEs. |
| **Pre-Strategy Intervention** | |
| $\mathcal{Z} \subset \mathcal{X}$ | The space of guidance signals representing all possible additional desired outcomes. |
| $Z \in \mathcal{Z}$ | A specific guidance signal representing an additional desired outcome. |
| $D_{pre}$ | Pre-strategy decision variable. |
| $\sigma_{pre}$ | A pre-strategy assigned to the pre-decision variable $D_{pre}$. |
| $\delta_{pre}$ | A pre-policy, which is a function that determines a pre-strategy $\sigma_{pre}$. |
| $\boldsymbol{\hat{\sigma}}_{\mathcal{I}}$ | The reduced set of NEs induced by an intervention. |
| **Utility, Probability Distribution and Causal Effect** | |
| $U_{team}$ | Team utility. |
| $U_{tot}$ | Total utility, representing the overall objective. |
| $U_{task}, U_{sec}$ | Utility of the primary task goal and the secondary goal, respectively. |
| $\mathbb{E}_{U_i}(\sigma)$ | Expected utility for agent $i$ under strategy profile $\sigma$. |
| $P_{\mathcal{M}[\sigma]}(U = u)$ | Likelihood of outcome $U = u$ under strategy profile $\sigma$. |
| $P_\sigma(\sigma)$ | Probability distribution over strategy profiles. |
| $P(\cdot \mid \text{do}(\sigma_{pre}))$ | Probability distribution after applying the causal intervention $\text{do}(\sigma_{pre})$. |
| $\Delta_{\text{CE}}^{\sigma_{pre}}(U = u)$ | Causal effect of a pre-strategy intervention on an outcome. |
| **MARL** | |
| $\mathcal{S}$ | Set of states in a Markov game. |
| $\mathcal{A}$ | Joint actions in a Markov game. |
| $T$ | Transition function in a Markov game. |
| $R$ | Team reward function in a Markov game. |
| $\pi$ | Joint policy of agents in a Markov game. |

# B   Related Work

## B.1   Multi-agent Team with Human Feedback

Our work connects to the growing field of leveraging human guidance within multi-agent systems [43]. A prominent related area is reinforcement learning from human feedback (RLHF), where human input, such as preferences or demonstrations, guides agent policy optimization [44, 45], often proving essential for tasks with complex or subjective goals for which explicit reward design is difficult [45, 46, 47]. RLHF is a specific instance within the broader scope of human-instructed multi-agent systems, which utilize diverse inputs like direct commands [48], demonstrations [49, 50], policy shaping [46], or language instructions [51] to steer agent behaviours. Incorporating such human expertise can significantly enhance system performance [44, 52, 53, 54], interpretability [11, 51, 55, 56], and alignment [45, 57]. These methods typically treat human feedback as a data source to directly train an agent's policy, often in a model-free manner.

In contrast, our work introduces a structural approach grounded in MARL interaction paradigms. We use multi-agent influence diagrams (MAIDs) to formally model the strategic dependencies within

a MARL learning process. Based on this model, our targeted intervention paradigm is designed to provably maximize the causal effect on a desired outcome, which can be provided by human feedback. The core advantage of our framework is therefore its ability to provide a principled, analytical method for designing an intervention on a multi-agent system and predicting its system-level impact. This is a significant advantage in scenarios where the effects of global, heuristic feedback are hard to foresee.

## B.2 Goal-Conditioned MARL

Our work relates to goal-conditioned reinforcement learning (GCRL), where agents learn policies conditioned on achieving diverse goals, such as reaching specific states [58, 59], matching images [60], or following language instructions [61]. While concepts similar to subgoals like roles [27], skills [62], or subtasks [63] are used in MARL for task decomposition or coordination. The foundational approach in this area is the option framework in hierarchical reinforcement learning (HRL) [64], where "options" (temporally extended sub-policies) are used to achieve subgoals.

Our "additional desired outcome" can serve a similar function to a subgoal, though our framework interprets this concept more broadly. A traditional subgoal often refers to a specific intermediate state required for task completion. However, our composite desired outcome can also encompass beneficial patterns of behaviour that, while not directly related to the task completion, significantly improve the MARL learning performance and coordination. For example, in the game Hanabi, a team can achieve a score without adhering to a specific convention. However, adopting a convention (an additional desired outcome) makes teammates' actions more predictable, thereby enhancing coordination and improving overall performance.

This broader focus on guiding patterns of behaviour, rather than just reaching explicit, predefined subgoals, motivates our distinct approach. Our approach differs from these methods in two principled ways. First, our Pre-Strategy Intervention provides a continuous guidance signal that influences an agent's primitive actions at each timestep, rather than directly invoking a temporally abstract sub-policy (e.g. option) to make decision. Furthermore, our approach is designed to be particularly advantageous in scenarios where providing global guidance to all agents is impractical. Our core distinction lies in using a principled framework to analyze and apply a targeted intervention to a single agent.

## B.3 Intrinsic Reward Method

A common approach to realize guidance in MARL is through maximizing intrinsic rewards during learning. These internally generated signals can encourage diverse behaviours motivated by curiosity [65, 66], novelty [67, 68], social influence [7], empowerment [9], learned reward functions [8, 69] or utilizing prior knowledge from large language models [23, 51].

While our Pre-Strategy Intervention is implemented via an intrinsic reward, our core distinction lies in its purpose and scope. Unlike methods such as those in [8, 23], which applied intrinsic rewards to all agents to encourage emergent coordination, our Pre-Strategy Intervention strategically directs the signal only to a single targeted agent. Furthermore, standard intrinsic rewards are generally designed to solve local challenges like sparse rewards via using heuristic metrics like novelty. In contrast, our guidance signal is explicitly designed to solve a system-level problem: reaching a desired outcome for the entire multi-agent system.

To achieve this, we leverage Multi-Agent Influence Diagrams (MAIDs) and their corresponding relevance graphs to formally analyze the strategic dependencies between agents. This causal analysis is crucial as it allows us to move beyond simple heuristics. We then strategically design a guidance signal, which is implemented as an intrinsic reward for a single targeted agent. The specific objective of this signal is to maximize its causal effect on the composite desired outcome.

Therefore, our contribution is not merely the use of an intrinsic reward, but rather the principled framework that uses causal analysis to inform the design of such a signal.

## B.4 Environment and Mechanism Design

Our work relates to concepts from environment and mechanism design. Environment design typically involves structuring or modifying environmental configurations to guide agent behaviours towards

a targeted outcome [70, 71, 72]. In contrast, our approach does not directly reconfigure the static environment. Rather, it focuses on intervening on a targeted agent. Consequently, the behaviour of the intervened agent changes, altering the effective environment dynamics experienced by other agents. This allows us to shape emergent team behaviour by modifying agent interaction paradigms rather than fixed environmental rules.

Our work can also be viewed through the lens of mechanism design, which focuses on designing the "rules of games" such that a desired outcome is attained [73, 74]. Our proposed targeted intervention paradigm can be categorized into this context as a mechanism that is applied to specific agents designed to steer the MARL learning process towards equilibria that satisfy both the primary task goal and a specified additional desired outcome.

### B.5 Probabilistic Graphical Models for Multi-Agent Games

Graphical models based on Bayesian networks were introduced to represent dependencies, relevance, and relationships among variables in multi-agent games by Koller and Milch [17]. This formulation has since inspired advancements in both algorithmic innovations for solving games [75, 76] and empirical analysis [77]. More recently, Hammond et al. [78] incorporated causal graphical models into multi-agent reasoning, extending multi-agent influence diagrams (MAIDs) to the causal domain. Their work introduced the concept of pre-policy intervention, allowing for queries such as "What if some agents commit to a policy before others make their decisions?" This framework has become a foundational tool in subsequent research on theory of mind [79, 80], reinforcement learning [81], and causal modeling of agents [40]. To maintain consistency with the MAID framework, we rename "pre-policy intervention" to "pre-strategy intervention" in this paper.

Our work builds directly on the causal framework proposed by Hammond et al. [78]. However, whereas their work introduced pre-policy intervention as a general tool for causal reasoning in game theory, our work is the first to adapt and apply this concept to solve a core challenge in MARL: achieving coordination by guiding a multi-agent system towards a desired outcome. We leverage MAIDs not just for analysis, but as a practical design tool to construct an MARL interaction paradigm. We design Pre-Strategy Intervention as the core mechanism within the targeted intervention paradigm, using it to guide the MARL learning process towards a desired, high-performing outcome.

## C  Extra Background

### C.1  An Example of MAIDs

We exemplify MAIDs through a two-agent scenario from [17].

**Example.**  Alice is considering building a patio behind her house, which would be more valuable if she could have a clear view of the ocean. However, a tree in the yard of her neighbor Bob blocks her view. Alice, being somewhat unscrupulous, contemplates poisoning Bob's tree, which would cost her some effort but might cause the tree to become sick. Bob is unaware of Alice's actions, but can observe if the tree starts to deteriorate, and he has the option of hiring a tree doctor (at a cost). The attention of the tree doctor reduces the chance that the tree will die during winter. Meanwhile, Alice must decide whether to build her patio before the weather turns cold. At the time of her decision, Alice knows whether a tree doctor has been hired but cannot directly observe the tree's health. An MAID for this scenario is shown in Figure 6.

In this example, *Poison Tree* and *Tree Doctor* are **s-reachable** to *Build Patio*, as Alice must take into account whether she has already poisoned the tree and whether Bob has called a tree doctor before deciding whether to build the patio.

### C.2  Additional Definitions

**Definition C.1** (**Active Path** [82]).  Let $\mathcal{G}$ be a Bayesian Network (BN) structure, and let $X_1 - X_2 - \cdots - X_n$ represent an undirected path in $\mathcal{G}$. Let $E$ be a subset of variables in $\mathcal{G}$ (the evidence set). The path $X_1 - \cdots - X_n$ is *active* given evidence $E$ if the following conditions hold:

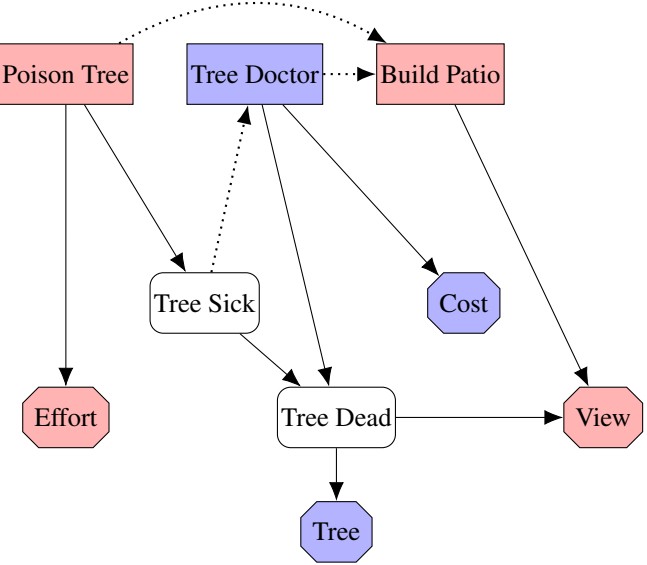

Figure 6: A MAID for the Tree Killer example; Alice's decision and utility variables are in red, and Bob's are in blue. Decision nodes are rectangular, chance nodes are squircular, and utility nodes are hexagonal.

1. Whenever a collider is on the path, i.e., a structure $X_{i-1} \to X_i \leftarrow X_{i+1}$, then either $X_i$ or one of its descendants is in $E$;

2. No other variables along the path is in $E$.

# D  Illustrative Example: Pre-strategy Intervention in a Logistics Game

**Background**  Two logistics companies, Company A and Company B, share a warehouse and use robots to manage inventory. Each company has two options:

- Optimize space usage: Focus on efficient organization.
- Prioritize speed: Focus on moving items quickly.

Both companies' choices affect each other's performance, and they aim to achieve the best outcome for their operations.

**Utility Table**

| Company A\Company B | Optimize Space Usage (B) | Prioritize Speed (B) |
|---|---|---|
| Optimize Space Usage (A) | $(9, 9)$ | $(3, 6)$ |
| Prioritize Speed (A) | $(6, 3)$ | $(5, 5)$ |

## D.1  Pre-Strategy Intervention

Our Pre-Strategy Intervention is applied to Company A (the targeted agent) to guide the system towards the more desired (9,9) equilibrium. The pre-policy module learns to provide a guidance signal to Company A that effectively incentivizes it to choose "Optimize Space Usage." For instance, this guidance might be implemented as an additional input signal that favorably alters Company A's perceived utility or Q-values for choosing "Optimize Space Usage," making it the optimal choice for Company A given the intervention.

By influencing Company A to select "Optimize Space Usage," and assuming Company B best responds to this action (or also converges to "Optimize Space Usage" due to the system dynamics and its own objectives), the system is steered towards the (9,9) outcome. This targeted intervention

on Company A thus fosters a cooperative outcome and prevents convergence to the less efficient (5,5) equilibrium, demonstrating how Pre-Strategy Intervention can be used for equilibrium selection in a MARL setting.

# E    Theoretical Proof of Proposition 3.4

The proposition relies on the assumption that the probability function $P_\mathcal{I}$ is upper semicontinuous. This is a reasonable assumption because the best-response function in a game is **not always** continuous. An intuitive example to support this assumption is the game *paper, rock, scissors*, where the best response is conducting each action uniformly. If we consider a pre-strategy that shifts one player towards slightly less likely playing rock, then the probability of the opponent playing paper would experience a "jump" to 0, which can be seen as a discontinuity in the function. A detailed example solution of pre-strategy intervention can be found in Appendix D.

*Proof.* Proposition 3.4 claims that an intervention exists: (a) it does not decrease the probability of the desired outcome and (b) a maximum-effect intervention exists.

(a) A trivial case exists where a pre-strategy that equals the marginal conditional probability of $U = u$ can be achieved by doing empty intervention.

(b) To prove that there exists a pre-strategy maximizing the causal effect, we observe that the second term on the right-hand side of Eq. (2) is constant. Therefore, maximizing the first term is equivalent to maximizing the causal effect.

To analyze how an agent's strategy influences the final outcome, we can decompose the probability of achieving the desired outcome $U_{tot} = u^*$. Using the law of total probability and the Markov property of the Bayesian network [17], we can marginalize out the decision variable $D$ and its parents $Pa(D)$. This decomposition explicitly isolates the agent's decision rule, $P_{\mathcal{M}[\sigma]}(d \mid \mathbf{pa}_D)$, which is the component that our intervention targets:

$$P_{\mathcal{M}[\sigma]}(U_{tot} = u^*) = \int_{\mathbf{pa}_D \in \mathrm{dom}(Pa(D))} P_{\mathcal{M}[\sigma]}(\mathbf{pa}_D) \, d\mathbf{pa}_D$$

$$\times \int_{d \in \mathrm{dom}(D)} P_{\mathcal{M}[\sigma]}(d \mid \mathbf{pa}_D) \, dd$$

$$\times P_{\mathcal{M}[\sigma]}(U_{tot} = u^* \mid d, \mathbf{pa}_D) \tag{4}$$

The function $f(\sigma_{pre})$, representing the expected probability of $U = u$ under the pre-strategy, is defined as:

$$f(\sigma_{pre}) := P_\mathcal{I}(U_{tot} = u^*) = \int_{\hat{\sigma} \in \hat{\boldsymbol{\sigma}}_\mathcal{I}} P_{\mathcal{M}[\hat{\sigma}]}(U_{tot} = u^*) P_{\hat{\sigma}}(\hat{\sigma}) \, d\hat{\sigma}$$

Assuming $f$ is an upper semicontinuous function defined on a compact domain $\mathrm{dom}(\sigma_{pre}) \subseteq \mathbb{R}^m$, we aim to demonstrate that $f$ has a maximum on this domain. This follows from the Extreme Value Theorem. We may replace the notation $\sigma_{pre}$ with $\sigma$ in the following steps for simplicity, with a slight abuse of notation.

**Boundedness Above**: Suppose, for contradiction, that $f$ is unbounded above. For each $k \in \mathbb{N}$, there exists $\sigma_k \in \mathrm{dom}(\sigma)$ such that $f(\sigma_k) > k$. Since $\mathrm{dom}(\sigma)$ is compact, the sequence $\{\sigma_k\}$ contains a convergent subsequence $\{\sigma_{k_l}\}$ converging to some $\sigma_0 \in \mathrm{dom}(\sigma)$.

The property of upper semicontinuity implies $\limsup_{l \to \infty} f(\sigma_{k_l}) \leq f(\sigma_0)$, which contradicts the assumption because it suggests $\limsup_{l \to \infty} f(\sigma_{k_l}) = \infty$. This shows f is bounded above. Then we can define:

$$\gamma = \sup\{f(\sigma) : \sigma \in \mathrm{dom}(\sigma)\}$$

Since the set $\{f(\sigma) : \sigma \in \mathrm{dom}(\sigma)\}$ is nonempty and bounded above, $\gamma \in \mathbb{R}$.

**Existence of Maximum**: Let $\{\sigma_k\}$ be a sequence in $\mathrm{dom}(\sigma$ such that $\{f(\sigma_k)\}$ converges to $\gamma$. By the compactness of the domain, the sequence $\{\sigma_k\}$ has a convergent subsequence $\{\sigma_{k_\ell}\}$ that converges to some $\bar{\sigma} \in \mathrm{dom}(\sigma)$. Then

$$\gamma = \lim_{\ell \to \infty} f(\sigma_{k_\ell}) = \limsup_{\ell \to \infty} f(\sigma_{k_\ell}) \leq f(\bar{\sigma}) \leq \gamma$$

**Conclusion**: The equality $\gamma = f(\bar{\sigma})$ establishes that $\gamma$ is the maximum value of $f$ on $\mathrm{dom}(\sigma)$, and thus $f(\sigma) \leq f(\bar{\sigma})$ for all $\sigma$ in the domain $\mathrm{dom}(\sigma)$.

$\square$

## F  Multi-Agent Influence Diagrams and Relevance Graphs in Implementation

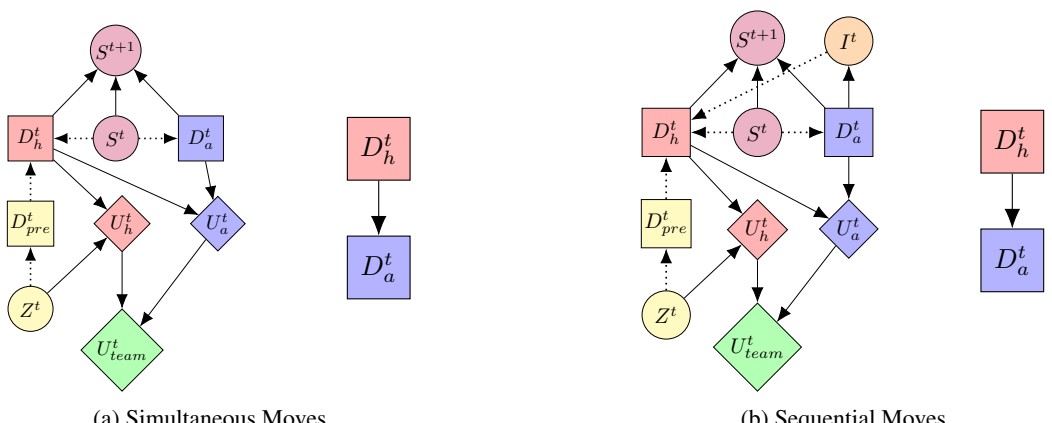

(a) Simultaneous Moves.                    (b) Sequential Moves.

Figure 7: MAIDs for pre-strategy intervention approach implemented in experiments, in sequential-move sequential decision makings, their corresponding Markov games and the relevance graphs for the Markov games. The figures illustrate the structure for (a) Simultaneous moves and (b) Sequential moves. The variable "$Z^t$" here indicates an intrinsic reward to implement an additional desired outcome, attributed to the the single targeted agent's individual utility variable $U_h^t$. This means the reachability of the additional desired outcome can be observed by the single targeted agent during learning.

In this section, we provide a more detailed illustration of the multi-agent influence diagram (MAID) structure that represents the practical implementation of our proposed Pre-Strategy Intervention. This shows how the intervention, designed to guide a single targeted agent towards an additional desired outcome, is integrated into that agent's decision-making process within the MAID framework. We present these implementation-specific MAIDs and their corresponding relevance graphs for sequential decision-making scenarios.

Figure 7a illustrates the MAID structures and its implementation for applying Pre-Strategy Intervention for sequential decision making with simultaneous moves. Similarly, Figure 7b shows these for sequential-move scenarios. These diagrams illustrate how the pre-decision variable ($D_{pre}^t$) and an additional desired outcome variable ($Z^t$) (represented as an intrinsic reward) directly inform the targeted agent's decision ($D_h^t$) at each relevant timestep. Furthermore, an edge from a variable representing the additional desired outcome $Z^t$ to the targeted agent's individual utility variable $U_h^t$ indicates that it can observe the reachability of the additional desired outcome during learning.

## G  Representing Team Reward Markov Games as Multi-Agent Influence Diagrams

This appendix details how a team reward Markov game can be equivalently represented as a Multi-Agent Influence Diagram (MAID).

Markov games [26] is a popular mathematical model to describe the multi-agent sequential decision process across various real-world applications [83, 39, 84]. For succinct description, we only consider the team reward Markov game with a finite episode length, as described in Definition G.1.

**Definition G.1 (Team Reward Markov Game [85]).** A team reward Markov game can be described as a tuple $\langle \mathcal{I}, \mathcal{S}, \mathcal{A}, T, R, L \rangle$. $\mathcal{I}$ is a set of agents; $\mathcal{S}$ is a set of states; $\mathcal{A} = \times_{i \in \mathcal{I}} \mathcal{A}_i$ is a set of joint actions and $\mathcal{A}_i$ is agent $i$'s action set; $T : \mathcal{S} \times \mathcal{A} \to \mathcal{S}$ describes the transition function that maps a state $s_t \in \mathcal{S}$ at timestep $t$ to $s_{t+1} \in \mathcal{S}$ at timestep $t + 1$; $R : \mathcal{S} \times \mathcal{A} \to \mathbb{R}$ is a team reward

function evaluating the immediate joint action $a_t \in \mathcal{A}$ at some state $s_t \in \mathcal{S}$. In a team reward Markov game with an episode length of $L$ timesteps, agents aim to learn a joint policy $\pi = (\pi_i)_{i \in \mathcal{I}}$ where $\pi_i : \mathcal{S} \to \mathcal{A}_i$ is agent $i$'s stationary policy, to solve the following optimization problem such that $\max_\pi \mathbb{E}_{\pi,T}[\sum_{t=0}^{L} R(s_t, a_t)]$.

A team reward Markov game can be represented as a MAID, because we can match variables between these two models. In more details, both models' agent sets are $\mathcal{I}$; $\mathcal{S}$ is associated with chance variables $\mathcal{X}$; $\mathcal{A}$ is associated with decision variables $\mathcal{D}$; $T$ is associated with conditional probability distributions $Pr$; $\pi$ is associated with decision rules $\delta$; and $\mathbb{E}_{\pi,T}[\sum_{t=0}^{L} R(s_t, a_t)]$ is associated with the expected utility shown in Eq. (1), with representing $R(s_t, a_t)$ as the team utility variable $U_{team}^t$.

For the self-organization, each agent's individual utility variable in the MAID can represent the equal contribution of the team utility variable $U_{team}^t$ that implies a team reward. According to [32], this is equivalent to the case where each agent's individual utility variable directly represents the shared team reward $R(s_t, a_t)$. Therefore, maximizing the objective of a team reward Markov game $\max_\pi \mathbb{E}_{\pi,T}[\sum_{t=0}^{L} R(s_t, a_t)]$ is equivalent to reaching a Nash equilibrium (NE) (Definition 2.2). For the targeted intervention, the team utility variable is defined as the total utility variable: $U_{team}^t := U_{tot}^t = U_{task}^t + U_{sec}^t$, where $U_{task}^t$ is shared among agents and $U_{sec}^t$ is only attributed to the targeted agent. Thus, the targeted agent's individual utility variable is defined as $U_{task}^t + U_{sec}^t$, while other agents' are defined as $U_{task}^t$. Therefore, maximizing each agent's individual utility leads to the preferred NE among multiple equilibria of the primary task utility. For the global intervention with an additional outcome, both $U_{task}^t$ and $U_{sec}^t$ are shared among agents. According to [32], each agent's individual utility variable is equally defined as $U_{task}^t + U_{sec}^t$. For the global intervention without an additional outcome, the situation is the same as the self-organization.

# H  Implementation Details

The implementation is mainly based on JaxMARL [86]. Our experiments were run on NVIDIA RTX 4090 and A100 GPUs. A run of experiments for the Hanabi environment typically completes in about one hour on these systems, while a run of experiments for the MPE environment only requires 5 minutes. The following paragraphs detail the architecture of our proposed method and the baselines used for comparison. Subsequently, we describe how the "additional desired outcome" is defined and implemented within the MPE and Hanabi environments respectively.

## H.1  Implementation of Our Method

Our method, **Pre-Strategy Intervention**, is realized by augmenting a base MARL agent's architecture with our novel **pre-policy module**. The full architecture is composed of three key components that work in concert: a standard network backbone, a Graph Neural Network (GNN) module for relational reasoning, and the pre-policy module for processing the intervention signal.

**Agent Architecture.** At each timestep, the agent's raw observation is processed in parallel by two components. First, the **GNN module** constructs a graph representation of the observation to capture relational features between entities (e.g., agents, landmarks), which aligns with the principles of MAIDs (full implementation details are provided in Appendix H.3). Second, the **pre-policy module** takes the same observation along with the intrinsic reward from the previous step to produce a guidance embedding. Finally, the outputs from the standard backbone, the GNN embedding, and the pre-policy guidance embedding are concatenated and fed into the agent's decision-making head (e.g., to compute Q-values or inform the critic).

This entire architecture, including the agent's policy, the GNN module and the pre-policy module, is trained jointly end-to-end to maximize a composite shaping reward. The composite shaping reward is the sum of the extrinsic task reward and the intrinsic reward derived from the additional desired outcome. As established in Section 4.3, this training process is theoretically sound, as maximizing the intervention's causal effect requires observing the total utility variable ($U_{tot}$), which directly aligns with the use of a shared team reward in many previous MARL works.

**Intrinsic Reward Function.** The intrinsic reward quantifies the agent's adherence to the specified "additional desired outcome" at each timestep. For example, in MPE this is the negative distance to a target landmark, while in Hanabi it is a value indicating compliance with a convention like "5 Save".

---

**Algorithm 1** Graph Embedding using GNNs

---

    **procedure** GRAPHEMBEDDING(Observation)
        **Input:** Observation vector
        Represent the observation as an influence diagram with semantic features
        Apply graph convolution using a GNN with an adjacency matrix (either learned or manually crafted)
        **Return:** Graph embedding vector
    **end procedure**

---

This intrinsic reward serves two purposes: (1) it is a direct input to the pre-policy module, and (2) it is added to the extrinsic task reward to form a composite shaping reward that guides the learning of the entire system.

**Architectural Choice (Non-Parameter Sharing).** Since our method focuses on the effect of intervening on a single targeted agent, other agents' parameters update should be isolated from the targeted agent. For this reason, it is most naturally implemented with a non-parameter sharing (NP) architecture. The Pre-Strategy Intervention is designed to influence only the targeted agent, and an NP setup allows for this specialized learning without forcing homogeneity across all agents' policies.

**Targeted Agent Selection.** For the experiments presented in this paper, we manually select a single targeted agent in each environment. This experimental design allows for a clear and controlled analysis of the mechanism's impact and mirrors practical scenarios where a human operator might manually choose a specific agent to intervene on.

## H.2 Implementation of Baselines

**Base MARL.** This refers to the standard, unmodified ("vanilla") version of the underlying MARL algorithm. For MPE, these include IQL [33], VDN [34] and QMIX [35]. For Hanabi, we use IPPO [36], MAPPO [37], and variants of PQN [38], specifically PQN-IQL and PQN-VDN. All Base MARL variants use the same network architecture and hyperparameters as our method but exclude both the pre-policy module and the additional intrinsic reward to maximize.

**Intrinsic Reward.** This approach serves as a direct ablation of our method to assess the contribution of the pre-policy module. In this variant, the pre-policy module is removed, while the additional desired outcome is conveyed to the single targeted agent implicitly via maximizing an intrinsic reward along with maximizing the extrinsic task reward.

**Global Intervention.** This approach represents a category of algorithms implementing the global intervention paradigm. In our experiments, we follow the specific implementation of two representative methods for this approach: LIIR [8] for policy-based algorithms, and LAIES [29] for value-based algorithms. Both methods in principle allocate an additional intrinsic reward to each agent. This contrasts the Intrinsic Reward approach above, where **only one** targeted agent is provided with an intrinsic reward to maximize. For LAIES, the intrinsic reward is implemented by a function calculating the causal effect of their actions on external states. The LAIES framework proposes two types of intrinsic motivation: Individual Diligence (IDI) and Collaborative Diligence (CDI). Notably, the training process for the CDI component relies on access to the joint actions of all agents. As our proposed method and the other baselines in our experiments operate without leveraging joint action information during training, we implemented the LAIES baseline using solely its IDI component to maintain consistency and ensure a fair comparison.

## H.3 GNN Implementation

The foundation of our targeted intervention method (Pre-Strategy Intervention) is formed by the s-reachability graph criterion, introduced to identify the relevant policies of other agents for decision-making [17]. However, considering only policies is insufficient, as an agent's policy may depend on other elements within the game [78]. Algorithm 1 provides an implementation how we can build connection with a causal graph structure for pre-policy learning. By leveraging this graph structure, we incorporate prior knowledge about the game to help guide agents' policy-making in practical. The feasibility of learning the causal graph during training has been demonstrated by

Richens and Everitt [87], where agents can learn the causal model implicitly during interaction with the environment.

**Architecture.** If the adjacency matrix is not predefined, the GNN processes the observation vectors by first encoding them into logits, which are used to generate a soft adjacency matrix via the gumbel-softmax technique [88]. This matrix defines the relationship between features in the observations. Once the adjacency matrix is formed, a graph convolutional layer applies message passing to update the features of each node based on its neighbors [82]. The output node features are then aggregated using a mean-pooling operation to produce a graph embedding. This embedding is used for further processing or decision-making.

## H.4 MPE

In the MPE Simple Spread task [19], our setup features 3 agents in a 2D continuous space cooperatively navigating to 3 landmarks while avoiding collisions. We apply our targeted intervention to one agent, using an intrinsic reward defined by the negative distance to its selected landmark. This is a simultaneous-move environment.

### H.4.1 Intrinsic reward

In the environment, we define two scenarios for providing intrinsic rewards to the targeted agent.

**Scenario 1 (Fixed Target Landmark).** The targeted agent receives an intrinsic reward for approaching a specific, predetermined landmark. This is implemented by treating one landmark (e.g., the first one listed in the observation, regardless of its random initial position) as the target. The agent receives an additional reward inversely proportional to its distance to this fixed target landmark.

**Scenario 2 (Dynamic Target Landmark — Farthest from Teammates).** The targeted agent receives an intrinsic reward for approaching the landmark currently farthest from its other two teammates. This encourages strategic positioning relative to teammates. The farthest landmark $\mathbf{l}^*$ is determined dynamically at each step as follows:

**Definition of the Farthest Landmark.** For each landmark $\mathbf{l} \in \mathcal{L}$, compute the Euclidean distances to teammates $\mathbf{p}_1$ and $\mathbf{p}_2$:

$$d_i(\mathbf{l}) = \|\mathbf{l} - \mathbf{p}_i\|_2, \quad \text{for } i = 1, 2.$$

Next, determine the minimum distance for each landmark:

$$d_{\min}(\mathbf{l}) = \min\{d_1(\mathbf{l}), d_2(\mathbf{l})\}.$$

Identify the farthest landmark $\mathbf{l}^*$ with the maximum minimum distance:

$$\mathbf{l}^* = \arg\max_{\mathbf{l} \in \mathcal{L}} d_{\min}(\mathbf{l}).$$

The intrinsic reward for the targeted agent is then calculated as:

$$\text{additional\_rew} = -\|\mathbf{p}_0 - \mathbf{l}^*\|_2,$$

where $\mathbf{p}_0$ is the targeted agent's position.

## H.5 Hanabi

Team success in the Hanabi card game heavily relies on shared *conventions*—implicit rules that enhance predictability and coordination [20]. Our experiments study the 2-agent version of the game. We investigate if our targeted intervention method (Pre-Strategy Intervention), applied to a single agent, can effectively promote adherence to such conventions and thereby improve overall team coordination. Specifically, we guide the targeted agent towards one of two common Hanabi conventions as a additional desired outcome: "5 Save" (prioritizing hints for unique, high-value cards to prevent accidental discards) or "The Chop" (making discards predictable by targeting the newest, unhinted card).

### H.5.1 Hanabi Environment and Experimental Setup Details

Hanabi is played by a team of 2-5 players (agents). The standard deck consists of cards in five different suits (colors), with cards in each suit numbered 1 to 5. There are three "1"s, two each of "2"s, "3"s, and "4"s, and one "5" for each suit, totaling 50 cards. The objective is for the team to collaboratively play cards in ascending order (1 through 5) for each of the five suits onto shared "firework" piles. Players hold a hand of cards (typically 4 or 5 cards, depending on the number of players), but crucially, they cannot see the cards in their own hand; they can only see the hands of their teammates.

On their turn, an agent must perform one of three actions:

- Give a Hint: The agent can give a hint to a teammate about the cards in that teammate's hand. To do so, they expend one of the team's shared "hint tokens". If no hint tokens are available, this action cannot be performed.
- Discard a Card: The agent selects a card from their own (unknown) hand to discard to a common discard pile. This action recover one hint token. The discarded card is revealed to all players and is then out of play. After discarding, the agent draws a new card from the deck.
- Play a Card: The agent selects a card from their own hand and attempts to play it onto one of the firework piles. If the card is playable (i.e., it is a "1" of a suit for which no card has been played, or it is the next number in sequence for a suit that has already been started), it is successfully added to the corresponding firework. If the card is not playable, it is discarded, and the team loses one of shared "lives". If all lives are lost, the game ends immediately, and the team scores 0. After playing (successfully or not), the agent draws a new card.

The team's score is the sum of the highest card values successfully played for each of the five suits.

The main difficulty in the Hanabi card game comes from the fact that players cannot see their own cards, a major limitation on what they know. Also, communication between players is very restricted and has a cost, as hints are limited. This requires players to make complicated deductions and makes working together effectively as a team extremely hard. For example, players need to discard cards to get more hint tokens. However, this is risky because they might accidentally remove unique and essential cards from the game, which could lower the team's possible score.

Consequently, high-performing Hanabi teams, both human and AI, heavily rely on shared conventions. These are implicit behavioural rules or patterns that enhance the predictability and informativeness of actions, particularly for hints and discards, beyond their literal meaning. However, establishing consistent and effective conventions across all agents is a known and significant hurdle for standard multi-agent reinforcement learning (MARL) approaches, which may struggle to ensure all agents converge to and adhere to the same implicit rules.

### H.5.2 Intrinsic Reward

In the environment, we implement two conventions for defining intrinsic rewards.

**Convention 1 (5 Save).** This convention is crucial for maximizing potential scores, as rank 5 cards are unique for each suit and essential for completing a firework. It aims to enhance team coordination by guiding agents to provide timely and clear hints about rank 5 cards held by teammates. This is especially important if those cards are currently unhinted and therefore at high risk of being accidentally discarded by a teammate unaware of their value. The objective is to ensure these vital cards are identified, preserved, and eventually played successfully. The specific logic for calculating the intrinsic reward, which encourages the targeted agent's convention-adherent hinting actions (e.g., when a teammate holds an unhinted rank 5 card), is detailed in Algorithm 2.

**Convention 2 (The Chop).** This convention promotes predictability in discarding actions, which in turn helps improve team efficiency and reduce miscoordination. It establishes a clear default discard target, often targeted as the newest card in a player's hand that has not yet received any hint (the "chop" card). By adhering to this rule, especially when no other plays or hints are clearly beneficial, an agent provides a predictable pattern for discards. This allows teammates to make more informed inferences about the game state and their own hands. The intrinsic reward encourages the targeted agent to follow this specific discard protocol. The detailed logic for identifying the "chop" card and calculating the reward based on the agent's discard action is provided in Algorithm 3.

---

**Algorithm 2** "5 Save" Convention

---

1: **procedure** CALCULATE 5 SAVE REWARD($a_t$, $T$, *hands*, hints)
2:     **Input:** action $a_t$, set of teammates $T$, hands *hands*, hint info hints.
3:     **Output:** Intrinsic reward $r_{5save}$.
4:     $r_{5save} \leftarrow 0$
5:     **if** any teammate $p \in T$ has an unhinted rank 5 card in *hand* **then**
6:         **if** $a_t$ is a hint action **and** $a_t$ does NOT hint rank "5" or the color of an unhinted 5 **then**
7:             $r_{5save} \leftarrow -1$                 ▷ Penalty: Failed to prioritize hinting the 5
8:         **end if**
9:     **end if**
10:    **Return:** $r_{5save}$
11: **end procedure**

---

---

**Algorithm 3** "The Chop" Convention

---

1: **procedure** CALCULATE THE CHOP REWARD($a_t$, hand, hints)
2:     **Input:** current action $a_t$, agent's hand *hand*, hint info *hints*.
3:     **Output:** Intrinsic reward $r_{chop}$.
4:     $r_{chop} \leftarrow 0$
5:     **if** $a_t$ is a discard action **then**
6:         Identify $chop\_card\_index$ using *hand* and *hints* (rightmost unhinted card index, or None if all hinted).
7:         Let $discarded\_index$ be the index discarded by $a_t$.
8:         **if** $chop\_card\_index$ is None **then**
9:             $r_{chop} \leftarrow -1$         ▷ Discouraged: Discarding when all cards are hinted
10:        **else if** $discarded\_index \neq chop\_card\_index$ **then**
11:            $r_{chop} \leftarrow -2$           ▷ Penalty: Discarded a non-chop card
12:        **end if**           ▷ Else (discarded the chop card): reward remains 0
13:     **end if**
14:    **Return:** $r_{chop}$
15: **end procedure**

---

# I   Additional Experiments

## I.1   MPE

### I.1.1   Detailed Results of the Scenario shown in the Main Paper

We now present the detailed results for the MPE scenario discussed in the main paper, comparing our targeted intervention approach (Pre-Strategy Intervention) against base MARL methods. These comparisons are conducted across various MARL algorithms, with results shown in Figure 8. The baseline methods are implemented in two common settings: parameter sharing (PS) and non-parameter sharing (NP). Our Pre-Strategy Intervention is implemented using a non-parameter sharing architecture, reflecting more realistic scenarios where agents may not be able to directly exchange parameters during training.

### I.1.2   Detailed Results of an Additional Scenario

In this scenario, the targeted agent receive additional feedback in intrinsic rewards, when it is close to the landmarks that are farthest from the another two teammates, as detailed in Appendix H.4. Figure 9 presents a detailed comparison of our targeted intervention approach (Pre-Strategy Intervention) against baseline methods, evaluated across various underlying MARL algorithms. Similarly, we consider baselines implemented in two popular implementation settings: parameter sharing (PS) and non-parameter sharing (NP). Our method, using non-parameter sharing, is observed to generally outperform the NP baselines by a significant margin in both primary task completion (extrinsic return) and attainment of the additional desired outcome (intrinsic return). Even compared with baselines implemented in PS, our method can still manifest comparable performance.

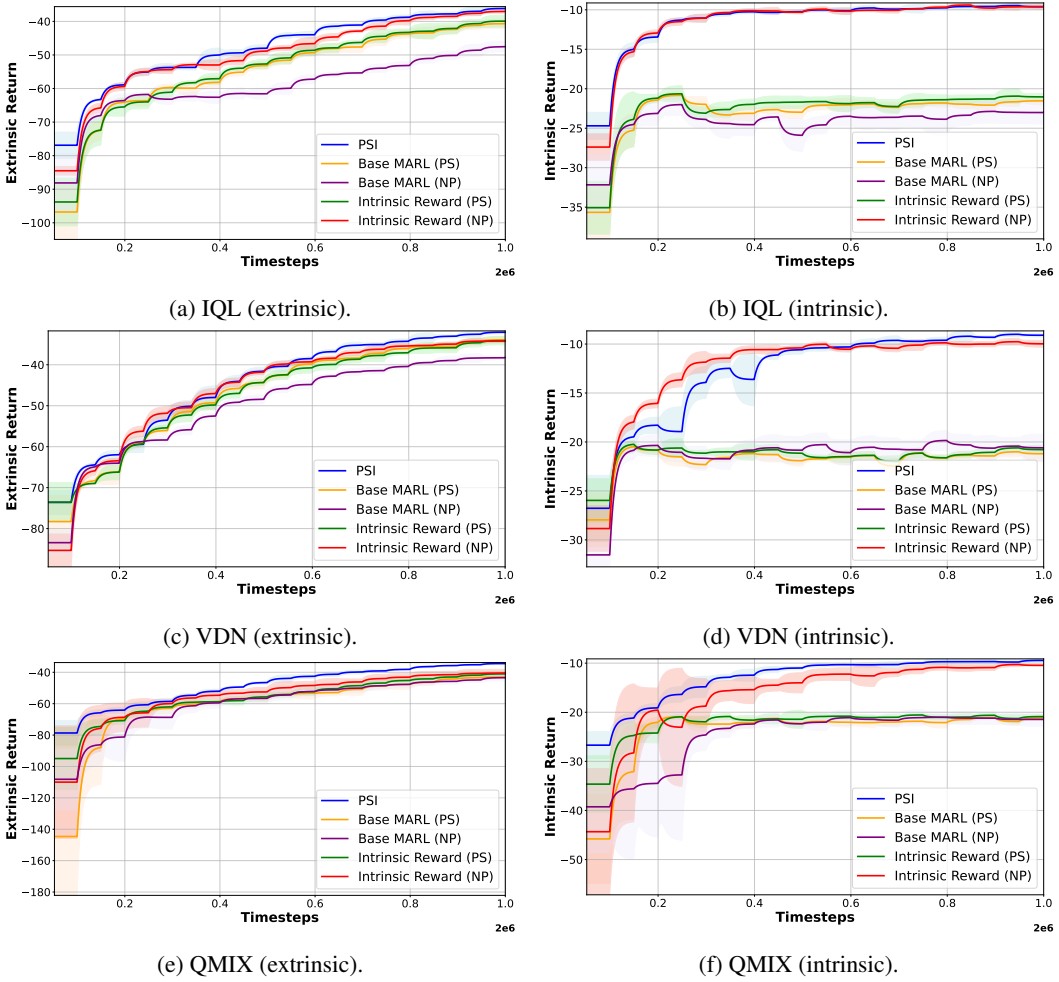

(a) IQL (extrinsic). (b) IQL (intrinsic).

(c) VDN (extrinsic). (d) VDN (intrinsic).

(e) QMIX (extrinsic). (f) QMIX (intrinsic).

Figure 8: Detailed results of the scenario of MPE shown in the main paper. For each base MARL algorithm (IQL, VDN, and QMIX), extrinsic and intrinsic returns are displayed side by side. The algorithm implemented in parameter sharing is denoted as PS, while that implemented in non-parameter sharing is denoted as NP.

## I.2 Hanabi

### I.2.1 Detailed Results of the Scenario shown in the Main Paper

We now present detailed results for the Hanabi scenario discussed in the main paper (using the "5 Save" convention as the additional desired outcome), comparing our targeted intervention approach (Pre-Strategy Intervention) against base MARL methods. These comparisons are conducted across various underlying MARL algorithms, with results shown in Figure 10. The baseline methods are implemented in two common settings: parameter sharing (PS) and non-parameter sharing (NP). Our Pre-Strategy Intervention is implemented using a non-parameter sharing architecture, reflecting more realistic scenarios where agents may operate with decentralized training and not directly exchange parameters during training.

### I.2.2 Detailed Results of an Additional Scenario

We now consider the scenario with a new convention called "The Chop." Similar to above, we present detailed results comparing our targeted intervention approach (Pre-Strategy Intervention) against baselines, across various underlying MARL algorithms, in Figure 11. The baselines again include implementations with parameter sharing (PS) and non-parameter sharing (NP). The results indicate that our method generally outperforms the non-parameter sharing (NP) baselines by a significant

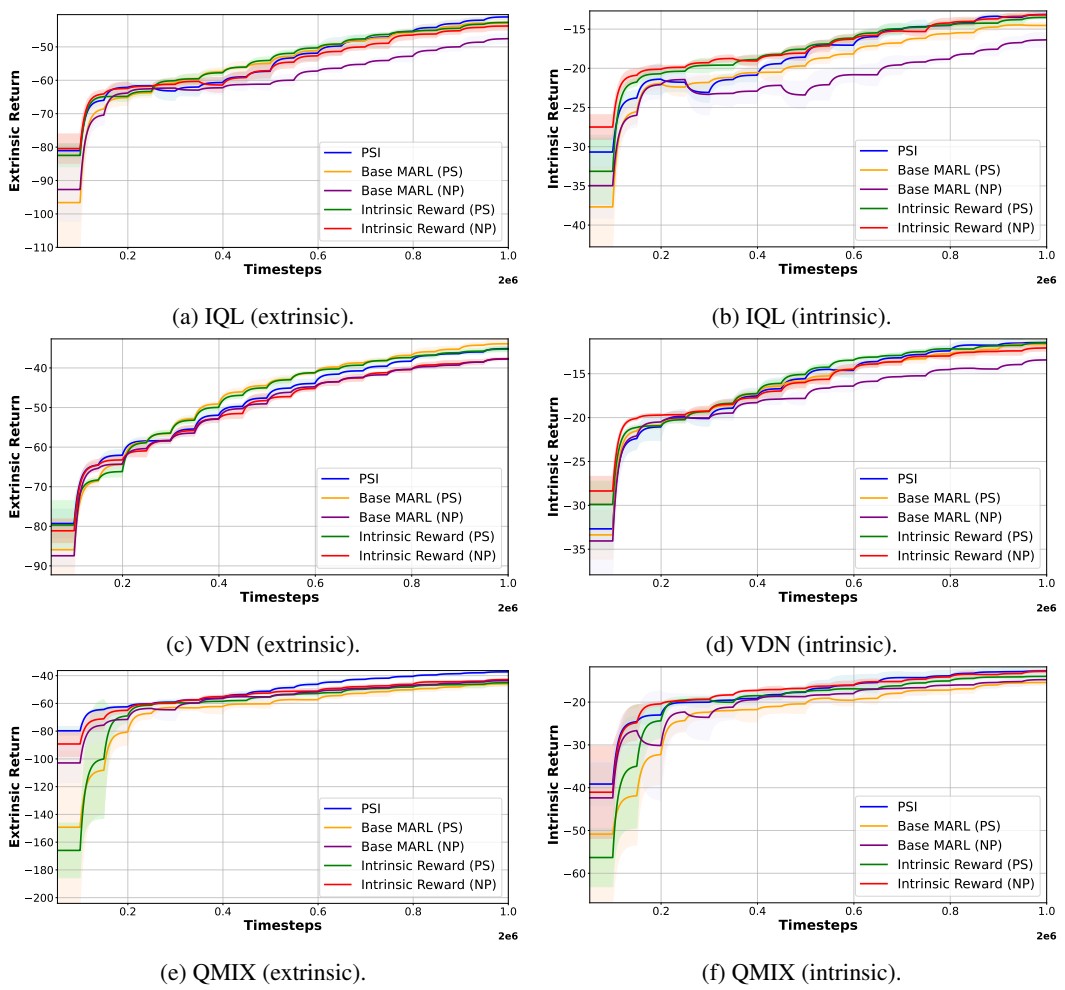

(a) IQL (extrinsic).      (b) IQL (intrinsic).

(c) VDN (extrinsic).      (d) VDN (intrinsic).

(e) QMIX (extrinsic).      (f) QMIX (intrinsic).

Figure 9: Detailed results of an additional scenario of MPE. For each base MARL algorithm (IQL, VDN, and QMIX), extrinsic and intrinsic returns are displayed side by side. The algorithm implemented in parameter sharing is denoted as PS, while that implemented in non-parameter sharing is denoted as NP. Our method is implemented in non-parameter sharing.

margin in both primary task completion (extrinsic return) and adherence to "The Chop" convention (intrinsic return). An exception is observed with the IPPO backbone, where the performance difference is less pronounced. Compared to parameter sharing (PS) baselines, our Pre-Strategy Intervention often demonstrates comparable performance.

### I.3 Additional Results Analysis

**Generalizability Across Different Additional Desired Outcomes and Base MARL Algorithms.** As shown in Figures 8, 9, 10, and 11, our targeted intervention approach (Pre-Strategy Intervention) consistently outperforms baseline approaches across various base MARL algorithms. Additionally, the results across different experimental scenarios (each with distinct additional desired outcomes, such as specific landmark targeting in MPE or convention adherence in Hanabi) suggest that incorporating well-defined additional desired outcomes via the Pre-Strategy Intervention can effectively promote the agent learning process towards improved task completion performance.

**Relationship Between Primary Task Goals and Additional Desired Outcomes.** We find that guiding the targeted agent (via the additional desired outcome) to move towards the landmark farthest from the other two teammates aligns with effective task completion in the MPE environment. As shown in Figure 9, the baselines exhibit an increasing trend in intrinsic reward compared to Figure 8,

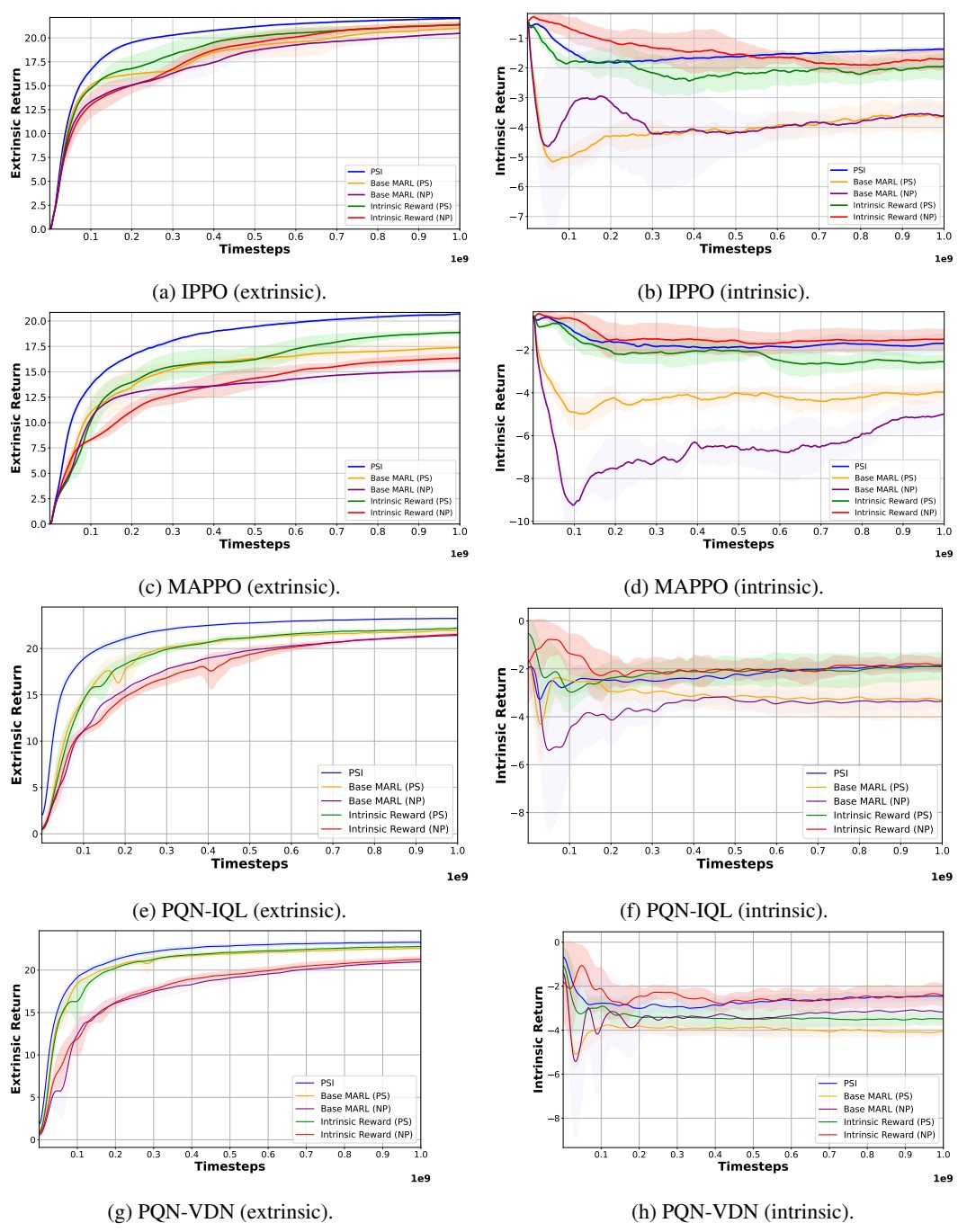

Figure 10: Detailed results of the scenario of Hanabi shown in the main paper (with the convention as "5 Save"). For each base MARL algorithm (IPPO, MAPPO, PQN-IQL and PQN-VDN), extrinsic and intrinsic returns are displayed side by side. The algorithm implemented in parameter sharing is denoted as PS, while that implemented in non-parameter sharing is denoted as NP. Our method is implemented in non-parameter sharing.

suggesting that moving towards the farthest landmark may share some overlap with the solution to the optimality of the task goal in MPE. However, in scenarios where the targeted agent's additional desired outcome is a randomly pre-assigned (fixed) landmark, the baselines do not show a similar spontaneous increase in intrinsic reward for that specific fixed landmark, as observed in Figure 8. This may be because moving to the farthest landmark is not inherently required to accomplish the primary

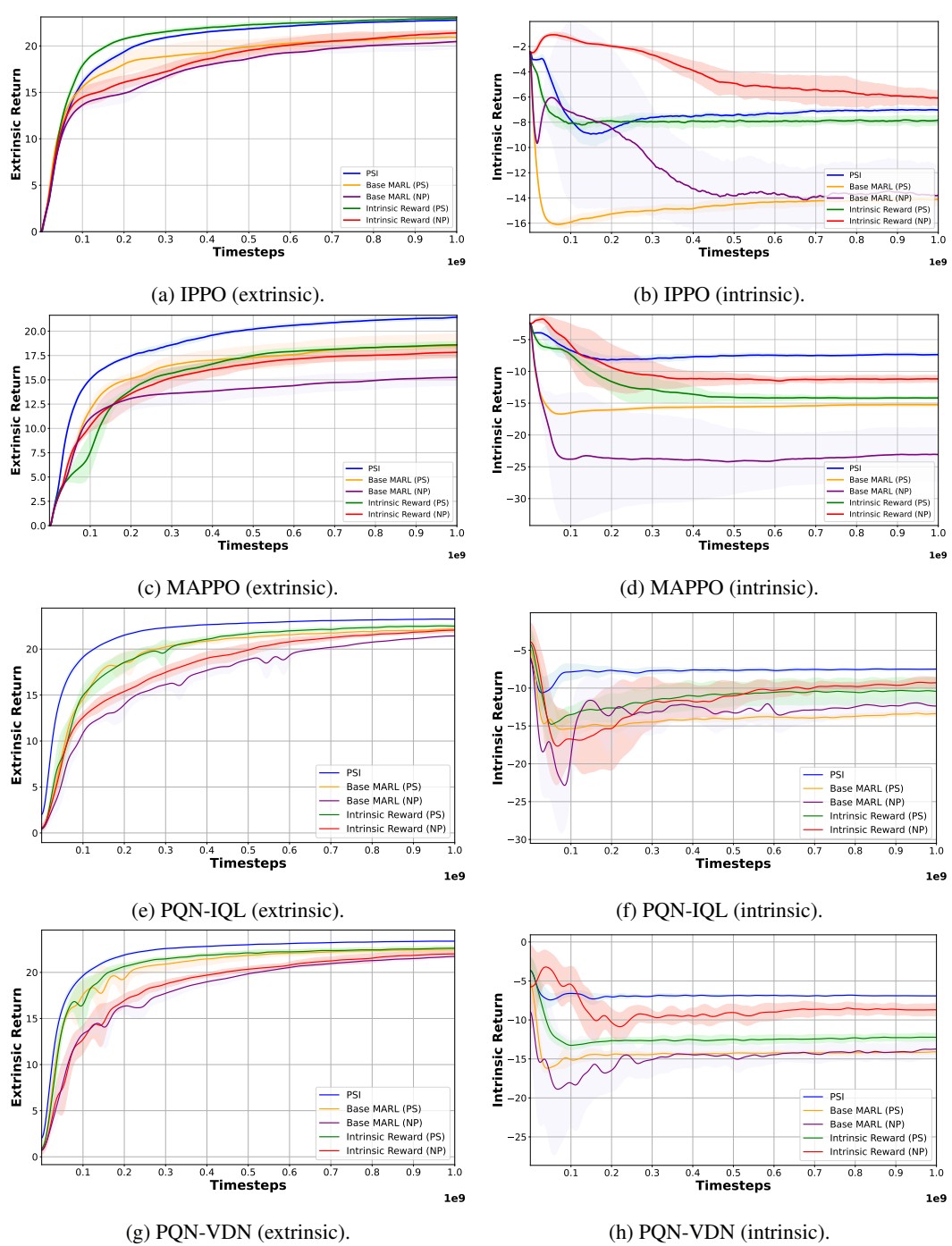

(a) IPPO (extrinsic).

(b) IPPO (intrinsic).

(c) MAPPO (extrinsic).

(d) MAPPO (intrinsic).

(e) PQN-IQL (extrinsic).

(f) PQN-IQL (intrinsic).

(g) PQN-VDN (extrinsic).

(h) PQN-VDN (intrinsic).

Figure 11: Detailed results of the scenario of Hanabi shown in the main paper (with the convention as "The Chop"). For each base MARL algorithm (IPPO, MAPPO, PQN-IQL and PQN-VDN), extrinsic and intrinsic returns are displayed side by side. The algorithm implemented in parameter sharing is denoted as PS, while that implemented in non-parameter sharing is denoted as NP. Our method is implemented in non-parameter sharing.

task goal. Instead, the predictable movement pattern of the targeted agent consistently favouring certain landmarks helps coordinate the team's movement, implicitly facilitating task completion.

## I.4  Additional Comparison with Additional-Outcome-Free Approaches in Hanabi

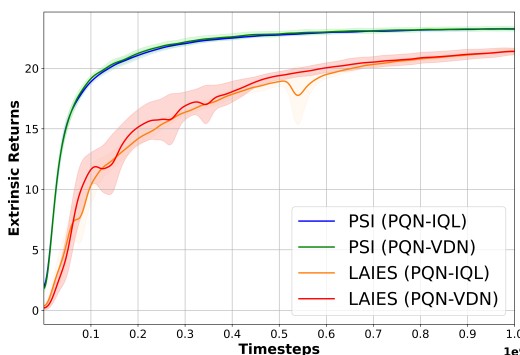

Figure 12: Comparison of extrinsic returns in Hanabi, evaluating our Pre-Strategy Intervention (PSI) implementing the targeted intervention paradigm against LAIES [29]) across various MARL algorithm backbones.

Figure 12 provides further evidence in the Hanabi environment, specifically utilizing value-based MARL algorithm backbones (PQN-VDN and PQN-IQL), for the comparison between our Pre-Strategy Intervention (PSI) implementing the targeted intervention paradigm and the LAIES [29] implementing the global intervention paradigm that only considers task completion. The results for extrinsic returns (task completion) are consistent with the findings presented in the main paper (e.g., Figure 5d, which used different backbones). When applied with either PQN-VDN or PQN-IQL, PSI generally leads to higher performance than LAIES.

## I.5  Experiment with Noisy Observation

To evaluate the robustness of our targeted intervention approach (Pre-Strategy Intervention) under **imperfect information** that is common in many real-world MARL settings, we introduce noise into the agents' observations. Specifically, we perturb the belief distributions agents form about cards in Hanabi, simulating scenarios with sensor noise or imperfect state estimation.

The experiment is conducted in Hanabi, using "The Chop" convention as the additional desired outcome guided by the intervention. For conciseness, we evaluate performance using the PQN-VDN backbone. We test the following three noise scenarios to assess robustness under different train-test conditions:

(1) **Noise during Training and Testing.** This simulates deploying the system in an environment with persistent observation noise, where the agent can potentially adapt during training.

(2) **Noise during Training Only.** This simulates training under noisy conditions but testing in a clearer environment. This tests robustness to a **decrease** in noise level compared to training conditions.

(3) **Noise during Testing Only.** This simulates training in relatively clean conditions but deploying in a noisy environment. This tests robustness to **unexpected** or higher levels of observation noise not seen during training.

### I.5.1  Both Training and Testing Containing Noise

In this scenario, the uniform noise (scaled by 0.2) perturbs the card belief distributions during both training and testing, simulating persistent observation uncertainty. As shown in Figures 13a and 13b, our method still generally outperforms the base MARL algorithms in primary task completion (extrinsic return). Furthermore, the adherence to the additional desired outcome (intrinsic return) remains effective, indicating robustness when agents can adapt to consistent noise during training.

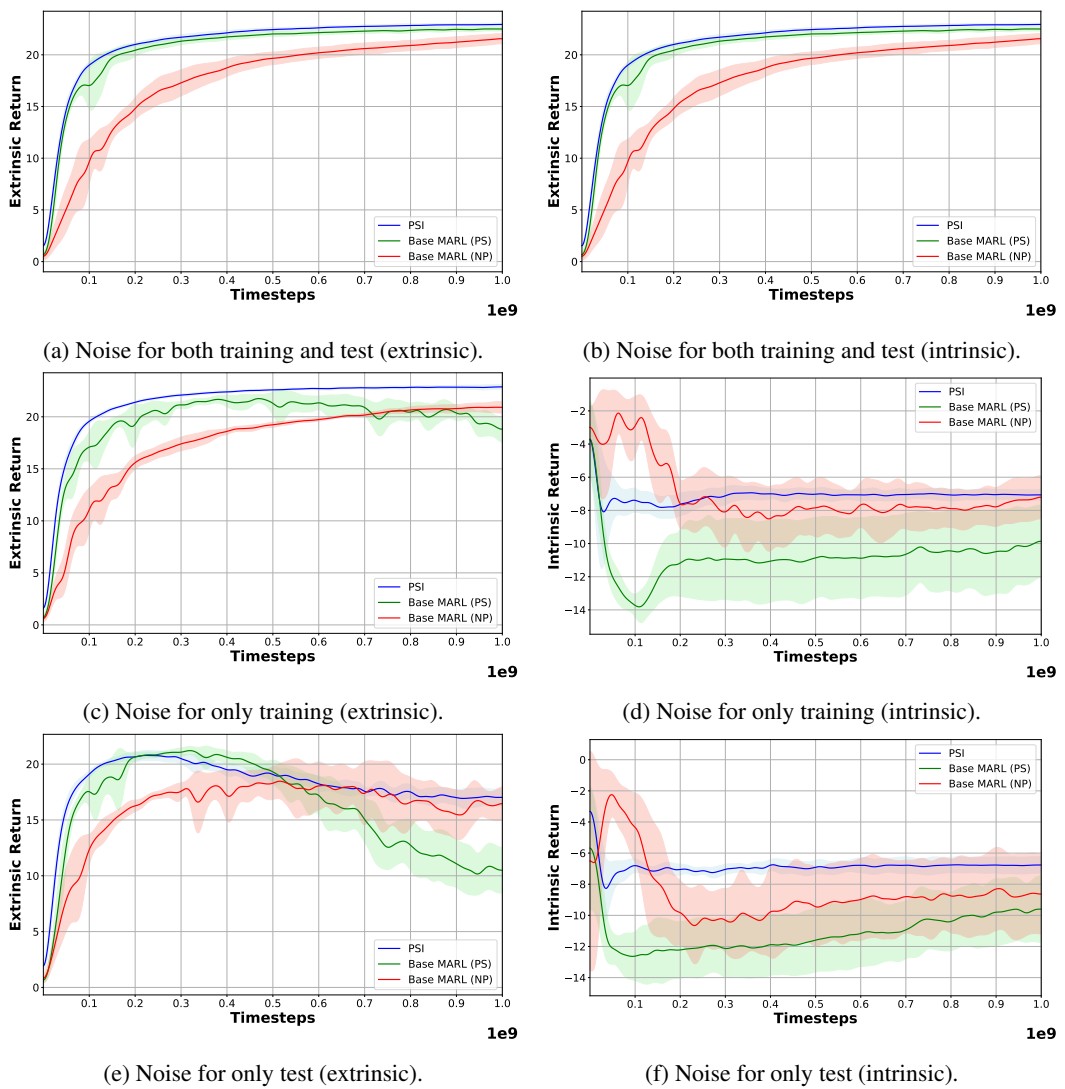

(a) Noise for both training and test (extrinsic).

(b) Noise for both training and test (intrinsic).

(c) Noise for only training (extrinsic).

(d) Noise for only training (intrinsic).

(e) Noise for only test (extrinsic).

(f) Noise for only test (intrinsic).

Figure 13: Comparison of performance across scenarios with different noise conditions imposed on training and testing. The first row shows results when both training and testing environments contain noise, the second row shows results when only the training environment contains noise, and the third row shows results when only the testing environment contains noise. For each scenario, extrinsic and intrinsic returns are displayed side by side. The algorithm implemented in parameter sharing is denoted as PS, while that implemented in non-parameter sharing is denoted as NP. Our method is implemented in non-parameter sharing. The base MARL algorithm we select is PQN-VDN.

### I.5.2 Only Training Environments Containing Noise

In this scenario, the noise (scaled by 0.05) is present only during training, simulating a scenario where the deployment environment has less noise than the training conditions. Figures 13c and 13d show that our method maintains an advantage in task completion over the baselines. However, the performance regarding the additional desired outcome (intrinsic return) is notably impaired compared to the noise-free or consistently noisy cases, performing similarly to the non-parameter sharing baseline.

### I.5.3 Only Testing Environments Containing Noise

This scenario introduces the noise (scaled by 0.02) only during testing, simulating deployment in an environment with unexpected observation noise. As seen in Figures 13e and 13f, the performance

of our method aligns closely with the parameter-sharing baseline in terms of extrinsic return (with overlapping confidence intervals). Similarly, there is no significant advantage in achieving the additional desired outcome (intrinsic return) compared to the non-parameter sharing baseline.

## I.6 Hanabi 4 Players

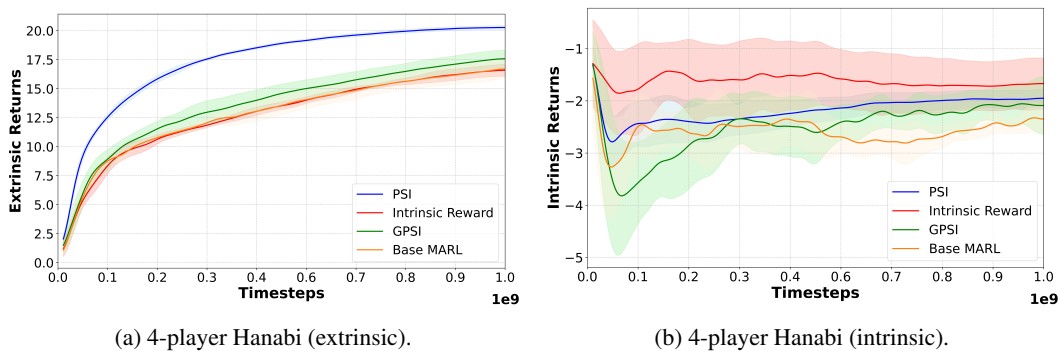

(a) 4-player Hanabi (extrinsic).

(b) 4-player Hanabi (intrinsic).

Figure 14: Performance comparison in 4-player Hanabi, with PQN-VDN as the base algorithm for all methods. Our Pre-Strategy Intervention (PSI) is compared against the Base MARL, the GPSI and the Intrinsic Reward baselines.

To further validate the scalability of our Pre-Strategy Intervention, we conducted new experiments on the 4-player version of the Hanabi challenge. This setting serves as a more demanding benchmark for coordination, as **the strategic complexity and communication burden are known to increase dramatically with the number of players** [89]. The results presented in Figure 14 demonstrate that our method achieves significantly higher scores than both the base MARL algorithms and the approach of the global intervention paradigm. This strong performance in a larger-scale setting provides robust evidence of our method's ability to effectively handle more complex coordination challenges.

## I.7 Heterogeneous MPE

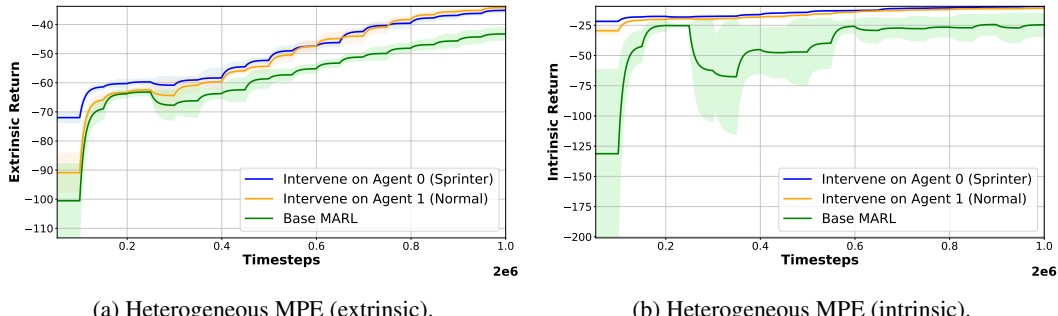

(a) Heterogeneous MPE (extrinsic).

(b) Heterogeneous MPE (intrinsic).

Figure 15: Performance evaluation in a Heterogeneous MPE scenario. We compare the effect of applying our Pre-Strategy Intervention to a fast agent versus a normal-speed agent, with IQL as the base algorithm, against a baseline without intervention (Base MARL). The heterogeneous environment consists of one fast and two normal-speed agents.

To evaluate the performance of our targeted intervention approach (Pre-Strategy Intervention) in settings with diverse agent capabilities, we designed a heterogeneous MPE scenario. This environment features three agents with asymmetric speeds: one "sprinter" agent with five times the normal speed and two normal-speed agents, with all methods built upon the IQL algorithm. The results, shown in Figure 15, reveal a consistent trend: intervening on a normal-speed agent leads to a noticeable improvement in final team performance (extrinsic return) compared to intervening on the uniquely capable sprinter, although the magnitude of this difference is modest in this specific environment. Our analysis suggests this is because the system's primary limitation: the **coordination bottleneck**

between the two normal-speed agents, when intervening on the sprinter agent. Even a subtle increase in the predictability of one normal agent's behaviour through our intervention can alleviate this core challenge, leading to a more effective overall team strategy and demonstrating our approach's potential in heterogeneous teams where targeted coordination adjustments can yield benefits.

## I.8 Extending Pre-Strategy Intervention to Global Intervention Paradigm

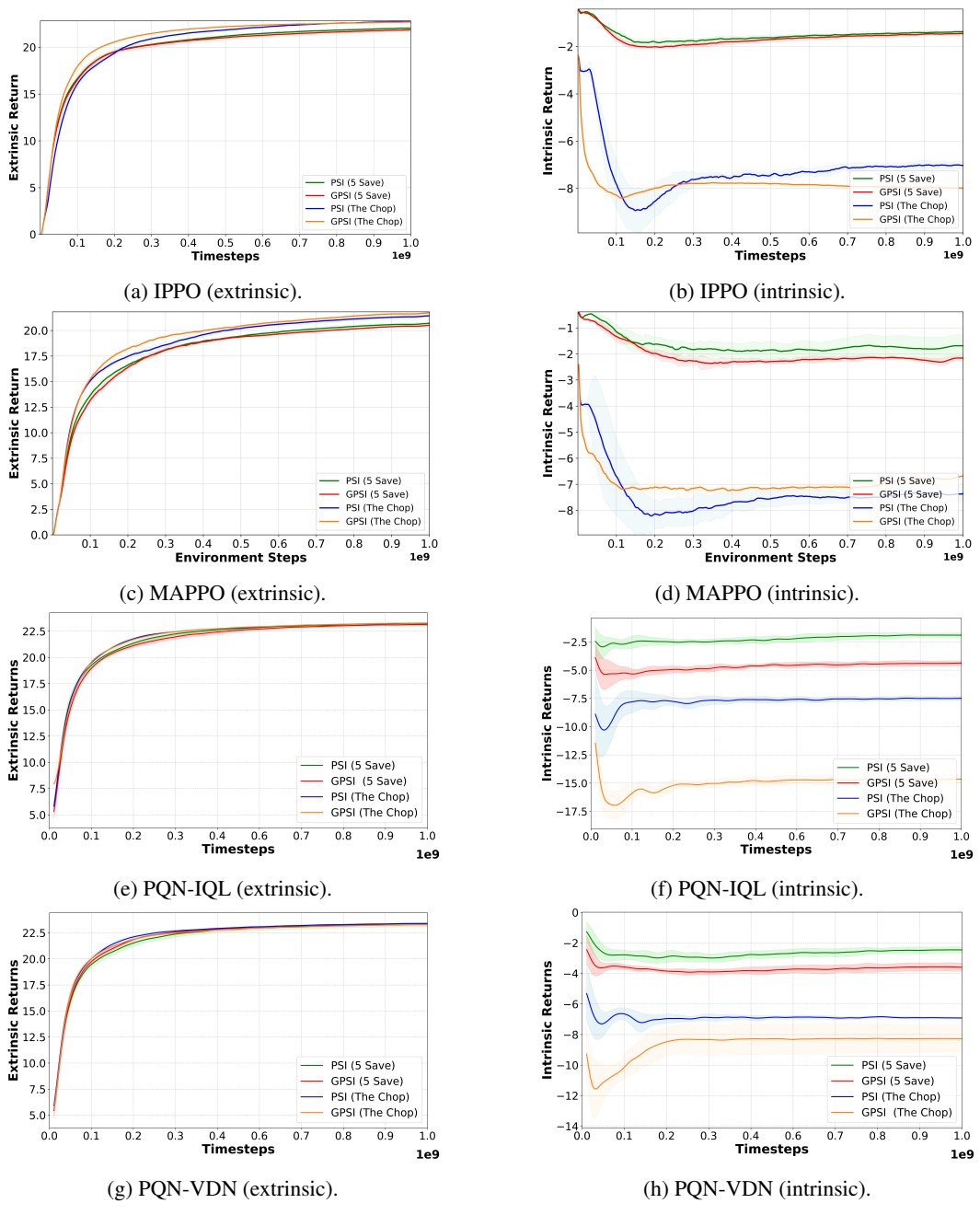

(a) IPPO (extrinsic).

(b) IPPO (intrinsic).

(c) MAPPO (extrinsic).

(d) MAPPO (intrinsic).

(e) PQN-IQL (extrinsic).

(f) PQN-IQL (intrinsic).

(g) PQN-VDN (extrinsic).

(h) PQN-VDN (intrinsic).

Figure 16: Impact of intervention scope on performance in Hanabi. While extrinsic returns (task completion) are similar, Targeted Intervention (on a single agent) leads to higher per-agent intrinsic returns (convention adherence) compared to a Global Intervention baseline where all agents are guided towards the convention. Intrinsic returns are averaged per intervened agent.

To understand the impact of intervention scope, Figure 16 compares our standard Targeted Pre-strategy Intervention (PSI) with the Global Pre-Strategy Intervention variant (GPSI) where the pre-policy module is applied to guide all agents simultaneously towards the convention. While the performance on primary task completion (extrinsic returns) is broadly similar between these two intervention scopes across the tested MARL algorithms, a notable difference appears in the intrinsic rewards (e.g., as shown in subfigures like 16b, 16f, and 16h for specific backbones). When PSI is targeted to a single agent, that agent typically achieves a higher intrinsic return (representing convention adherence) than the average per-agent intrinsic return obtained by GPSI. This suggests that targeted intervention may lead to more focused and efficient convention learning by the targeted agent, potentially avoiding redundancies or conflicting convention-based actions that can arise when all agents are simultaneously guided by the same pre-strategy.

## J  Broader Impacts

Our work on Pre-Strategy Intervention (implementing the targeted intervention paradigm) in multi-agent reinforcement learning (MARL) introduces a theoretical framework for agent coordination. As foundational research conducted entirely in simulated environments, the immediate societal impact is limited. However, we acknowledge that theoretical advances often lay groundwork for future applications. If extended to practical domains, this research could potentially enable more efficient coordination in multi-agent systems by focusing guidance on specific agents rather than requiring direct control of an entire system. Our targeted intervention paradigm could reduce computational and communication overhead in domains like robotic teams, traffic management, or resource allocation. At the same time, we recognize that techniques for influencing multi-agent systems through targeted intervention could potentially be misapplied if deployed in socially sensitive contexts without appropriate oversight. Future work that moves this theoretical foundation toward real-world applications will carefully consider application-specific ethical implications and incorporate appropriate safeguards against potential misuse.

## K  Hyperparameters

We show the hyperparameters for all experiments we conduct for the ease of reproducing results.

Table 2: Hyperparameters for MPE in IQL across two scenarios.

| Hyperparameter | Scenario 1 | Scenario 2 |
|---|---|---|
| Total Timesteps | $2 \times 10^6$ | $2 \times 10^6$ |
| Number of Environments | 8 | 8 |
| Number of Steps | 26 | 26 |
| Buffer Size | 5000 | 5000 |
| Buffer Batch Size | 32 | 32 |
| Network Hidden Size | 64 | 64 |
| Epsilon Start | 0.8 | 0.6 |
| Epsilon Finish | 0.02 | 0.1 |
| Epsilon Decay | 0.1 | 0.1 |
| Learning Rate (LR) | 0.0035 | 0.0045 |
| Learning Starts | 10000 timesteps | 10000 timesteps |
| Gamma ($\gamma$) | 0.9 | 0.9 |
| Max Gradient Norm | 5 | 5 |
| Target Update Interval | 200 | 200 |
| Tau ($\tau$) | 1.0 | 1.0 |
| Number of Epochs | 1 | 1 |
| Intrinsic Reward Ratio | 0.6 | 0.2 |
| Pre-Policy Module Output Size | 16 | 32 |
| Pre-Policy Module Hidden Size | 64 | 64 |
| GNN Node Embedding Size | 16 | 16 |

Table 3: Hyperparameters for MPE in VDN across two scenarios.

| Hyperparameter | Scenario 1 | Scenario 2 |
|---|---|---|
| Total Timesteps | $2 \times 10^6$ | $2 \times 10^6$ |
| Number of Environments | 8 | 8 |
| Number of Steps | 26 | 26 |
| Buffer Size | 5000 | 5000 |
| Buffer Batch Size | 32 | 32 |
| Network Hidden Size | 64 | 64 |
| Epsilon Start | 0.8 | 0.6 |
| Epsilon Finish | 0.1 | 0.1 |
| Epsilon Decay | 0.1 | 0.1 |
| Learning Rate (LR) | 0.007 | 0.0045 |
| Learning Starts | 10000 timesteps | 10000 timesteps |
| Gamma ($\gamma$) | 0.9 | 0.9 |
| Max Gradient Norm | 5 | 5 |
| Target Update Interval | 200 | 200 |
| Tau ($\tau$) | 1.0 | 1.0 |
| Number of Epochs | 1 | 1 |
| Intrinsic Reward Ratio | 0.6 | 0.2 |
| Pre-Policy Module Output Size | 32 | 32 |
| Pre-Policy Module Hidden Size | 64 | 64 |
| GNN Node Embedding Size | 16 | 16 |

Table 4: Hyperparameters for MPE in QMIX.

| Hyperparameter | Value |
|---|---|
| Total Timesteps | $2 \times 10^6$ |
| Number of Environments | 8 |
| Number of Steps | 26 |
| Buffer Size | 5000 |
| Buffer Batch Size | 32 |
| Network Hidden Size | 64 |
| Epsilon Start | 0.6 |
| Epsilon Finish | 0.1 |
| Epsilon Decay | 0.1 |
| Learning Rate (LR) | 0.004 |
| Learning Starts | 10000 timesteps |
| Gamma ($\gamma$) | 0.9 |
| Max Gradient Norm | 5 |
| Target Update Interval | 200 |
| Tau ($\tau$) | 1.0 |
| Number of Epochs | 1 |
| Intrinsic Reward Ratio | 0.2 |
| Pre-Policy Module Output Size | 32 |
| Pre-Policy Module Hidden Size | 64 |
| GNN Node Embedding Size | 8 |
| Mixer Embedding Size | 32 |
| Mixer Hypernetwork Hidden Size | 128 |
| Mixer Initialization Scale | 0.001 |

Table 5: Hyperparameters for Independent PPO in MPE.

| Hyperparameter | Value |
|---|---|
| Total Timesteps | $1 \times 10^{7}$ |
| Number of Environments | 16 |
| Number of Steps | 128 |
| Learning Rate | 0.00055 |
| Hidden Size | 64 |
| Update Epochs | 4 |
| Number of Minibatches | 8 |
| Gamma | 0.99 |
| GAE Lambda | 0.85 |
| Clip Epsilon | 0.2 |
| Scale Clip Epsilon | False |
| Entropy Coefficient | 0.015 |
| Value Function Coefficient | 0.95 |
| Max Gradient Norm | 10 |
| Activation | Tanh |
| Learning Rate Annealing | True |
| Intrinsic Reward Ratio | 0.2 |
| Pre-Policy Module Hidden Size | 64 |
| Pre-Policy Module Output Size | 16 |
| GNN Node Embedding Size | 8 |

Table 6: Hyperparameters for PQN in Hanabi.

| Hyperparameter | Value |
|---|---|
| Total Timesteps | $1 \times 10^{9}$ |
| Number of Environments | 1024 |
| Number of Steps | 1 |
| Hidden Dimension | 512 |
| Number of Layers | 3 |
| Normalization Type | Layer Normalization |
| Normalize Inputs | False |
| Dueling | True |
| Epsilon Start | 0.03 |
| Epsilon Finish | 0.005 |
| Epsilon Decay | 0.05 |
| Max Gradient Norm | 0.5 |
| Number of Minibatches | 1 |
| Number of Epochs | 1 |
| Learning Rate | 0.0002 |
| Learning Rate Linear Decay | True |
| Lambda | 0.0 |
| Gamma | 0.99 |
| Intrinsic Reward Ratio | 0.02 |
| Pre-Policy Module Hidden Size | 512 |
| Pre-Policy Module Output Size | 128 |
| GNN Node Embedding Size | 8 |
| GNN Observation Encoder Hidden Size | 64 |
| GNN Adjacency Matrix Sampling Temperature | 1.0 |

Table 7: Hyperparameters for IPPO in Hanabi.

| Hyperparameter | Value |
| --- | --- |
| Total Timesteps | $1 \times 10^9$ |
| Number of Environments | 1024 |
| Number of Steps | 128 |
| Learning Rate | 0.0006 |
| Anneal Learning Rate | False |
| Update Epochs | 8 |
| Number of Minibatches | 8 |
| Gamma | 0.99 |
| GAE Lambda | 0.9 |
| Clipping Epsilon | 0.1 |
| Entropy Coefficient | 0.02 |
| Value Function Coefficient | 0.5 |
| Max Gradient Norm | 0.5 |
| Intrinsic Reward Ratio | 0.035 |
| Pre-Policy Module Hidden Size | 256 |
| Pre-Policy Module Output Size | 128 |
| GNN Node Embedding Size | 8 |
| GNN Observation Encoder Hidden Size | 128 |
| GNN Adjacency Matrix Sampling Temperature | 1.0 |

Table 8: Hyperparameters for MAPPO in Hanabi.

| Hyperparameter | Value |
| --- | --- |
| Total Timesteps | $1 \times 10^9$ |
| Number of Environments | 1024 |
| Number of Steps | 128 |
| Learning Rate | 0.0065 |
| Anneal Learning Rate | True |
| Number of Update Epochs | 4 |
| Number of Minibatches | 4 |
| Gamma | 0.99 |
| GAE Lambda | 0.95 |
| Clipping Epsilon | 0.1 |
| Entropy Coefficient | 0.01 |
| Value Function Coefficient | 0.5 |
| Max Gradient Norm | 0.5 |
| Activation Function | ReLU |
| Intrinsic Reward Ratio | 0.035 |
| Pre-Policy Module Hidden Size | 128 |
| Pre-Policy Module Output Size | 256 |
| GNN Node Embedding Size | 8 |
| GNN Observation Encoder Hidden Size | 128 |
| GNN Adjacency Matrix Sampling Temperature | 1.0 |

