# OpenReview forum: "A Principle of Targeted Intervention for Multi-Agent Reinforcement Learning"
_NeurIPS.cc/2025/Conference — NeurIPS 2025 poster_

### Official Review · Reviewer_djxC · 2025-06-13

**Clarity:** 1
**Significance:** 2
**Originality:** 2
**Rating:** 2
**Confidence:** 3

**Summary:**

The authors introduce a Targeted Intervention methodology for improving system performance in multi-agent reinforcement learning (MARL) scenarios. In the Targeted Intervention methodology, a single agent is identified which is augmented with additional capabilities in order to facilitate more effective learning by all agents in the system. The authors compare their methodology against numerous baselines, including a simple intrinsic reward methodology, and show small but measurable improvements in system behavior.

**Questions:**

How many agents were simultaneously interacting in each of the evaluation scenarios (MPE and Hanabi)?

Given the additional perceptual abilities granted to agents, why is this paper focused on achieving a Nash equilibrium rather than seeking a far more desirable Pareto-optimal equilibrium?

Was a different set of baseline algorithms selected for each of the evaluation tasks (MPE and Hanabi)? The results would be far more illuminating if the same set of baselines was used in both sets of experiments.

**Ethical Concerns:**

["NO or VERY MINOR ethics concerns only"]

**Final Justification:**

I thank the authors for their responses.

While there are clearly interesting ideas explored in this paper, the limited scope of the experiments (2 agents in one instance, 3 in another) and the relatively small gains reported in the results section leave me unsatisfied with the work as a whole. I would be interested to see the application of these ideas in more complex scenarios.

As a stylistic note: I disliked the authors' repeated appeals to the opinions of other reviewers as a justification for the work. Strong work should be able to stand on its own, without appeals to external sources of validation.

**Limitations:**

This paper does not discuss the limitations of the work or identify potential areas of weakness. The authors are encouraged to consider the bounds within which their work is likely to be useful, and identify those boundaries for the edification of other researchers. It may also be worth identifying the limitations implied by seeking Nash equilibria rather than Pareto-optimal equilibria, and discussing the likely applicability (or lack thereof) of this methodology in real-world contexts where many aspects of the system cannot be known or observed.

**Paper Formatting Concerns:**

The Conclusion asserts: “We introduced Multi-Agent Influence Diagrams (MAIDs) as a principled framework for Designing and analyzing Targeted Interventions in MARL systems.” However, MAIDS is not a contribution of the current paper, having been introduced by Koller and Milch in 2003; only the application of MAIDS to the authors’ own Targeted Intervention methodology is novel, making the statement in the Conclusion misleading.

The acronym MPE is used on page 2, but is not defined until page 8.

Font sizes in Figure 5 are too small, and the line patterns/colors are not consistent across sub-plots, making it difficult to interpret and compare results for the various methods.

The following sentence contains a type: “Our approach first defines the pre-strateg intervention mechanism itself”

**Quality:**

2

**Strengths And Weaknesses:**

Strengths

- The authors provide complete technical detail and a formal description of the proposed methodology
- The authors compare their work against multiple baselines using both intrinsic and extrinsic reward measurements
- The authors are tackling an important problem with many real-world applications

Weaknesses

- The writing style over-emphasizes technical detail to the point that the main objectives of the work are difficult to understand. The authors are encouraged to provide simple layman's explanations of their methodology, while still maintaining the rigor of their formal descriptions
- The core contribution is minimal, and results in only minor performance improvements
- The Conclusion includes an ambiguous statement that gives the impression that the authors are the originators of the MAID framework, when in reality the authors are instead applying this framework within their specific use case

---

> ### Author Rebuttal · Authors · 2025-07-30
>
> # Response to Reviwer djxC
>
> We thank you for your detailed feedback. We appreciate them recognizing the technical completeness of our formal description, the thoroughness of our baseline comparisons, and the importance of the problem we are tackling.
>
> ### **Re: How many agents were interacting?**
> For the MPE experiments, we used 3 agents. For the Hanabi experiments, we used the standard 2-agent version, as detailed in Appendices H.2 and H.3.
>
>
> ### **Re:  Why is this paper focused on achieving a Nash equilibrium rather than seeking a far more desirable Pareto-optimal equilibrium?**
>
> Our work focuses on achieving a **specific, desirable system-level outcome** that is defined by a human designer. This outcome is a composite of the primary task goal and an additional desired property (e.g., adhering to a convention).
>
> The concept of Pareto optimality is centered on the individual utilities of the agents—a state is Pareto-optimal if no single agent can be made better off without making another agent worse off. **While this is a crucial concept for social welfare, it is not our direct objective.** The human-defined goal we aim for may not be a social optimum from the agents' perspective, and it's possible that not all agents individually benefit from the intervention in terms of their local rewards. Our evaluation is based on whether the system achieves the human-specified goal. Therefore, Nash Equilibrium (NE) is the more relevant solution concept for our work.
>
>
> ### **Re: Why were different sets of baseline algorithms selected?**
>
> We chose baselines that are considered standard and strong performers for each specific environment. This approach is common practice in the MARL community, as different classes of algorithms are often better suited to different types of coordination challenges. This is reflected in benchmark suites like JaxMARL [1], which often evaluate distinct sets of algorithms on different environments.
>
> To ensure a thorough and fair comparison, we evaluated our method against a wide range of algorithms. For instance, in **MPE**, in addition to the value-based methods reported in the paper (IQL, VDN, QMIX), we also added new experiments with the policy-based **IPPO**. The results below show a clear and significant performance improvement when our Targeted Intervention is applied.
>
> #### **IPPO MPE Results (Extrinsic Returns)**
>
> | Timesteps | 10% | 20% | 30% | 40% | 50% | 60% | 70% | 80% | 90% | 100% |
> | :--- | :---: | :---: | :---: | :---: | :---: | :---: | :---: | :---: | :---: | :---: |
> | **Baseline IPPO** | -64.84 | -54.03 | -50.41 | -48.74 | -46.60 | -45.17 | -43.57 | -42.85 | -42.88 | -43.32 |
> | **Our Method (IPPO)** | -39.28 | -36.24 | -33.25 | -32.24 | -31.29 | -31.77 | -30.98 | -30.91 | -29.82 | -29.85 |
>
> #### **IPPO MPE Results (Intrinsic Returns)**
>
> | Timesteps | 10% | 20% | 30% | 40% | 50% | 60% | 70% | 80% | 90% | 100% |
> | :--- | :---: | :---: | :---: | :---: | :---: | :---: | :---: | :---: | :---: | :---: |
> | **Baseline IPPO** | -23.48 | -21.70 | -21.49 | -21.22 | -21.37 | -21.39 | -21.29 | -20.19 | -20.81 | -20.80 |
> | **Our Method (IPPO)** | -10.15 | -9.59 | -9.12 | -8.98 | -8.27 | -8.66 | -8.64 | -8.56 | -8.92 | -8.54 |
>
> Similarly, for **Hanabi**, while the main paper highlights results against policy-based methods (IPPO, MAPPO), our appendix provides a full evaluation against strong value-based methods as well, specifically **PQN-IQL** and **PQN-VDN**. PQN is a popular and powerful baseline for Hanabi [2].
>
> By evaluating against both value-based and policy-based algorithms across our environments, we have ensured our comparisons are robust and tested against strong, modern implementations from both major classes of MARL algorithms.
>
>
> [1] Rutherford A, Ellis B, Gallici M, et al. Jaxmarl: Multi-agent rl environments and algorithms in jax[J]. Advances in Neural Information Processing Systems, 2024, 37: 50925-50951.
>
> [2] Gallici M, Fellows M, Ellis B, et al. Simplifying deep temporal difference learning[J]. arXiv preprint arXiv:2407.04811, 2024.
>
>
> ### **Re: This paper does not discuss the limitations of the work or identify potential areas of weakness.**
>
> We **respectfully disagree with assessment that the paper does not discuss limitations.** We would like to gently point the you to **Section 6, "Limitation and Future Work,"** where we explicitly address the primary limitations of our current framework.
>
> In this section, we directly address the exact point you raises about the "likely applicability (or lack thereof) of this methodology in real-world contexts where many aspects of the system cannot be known or observed." We state that:
>
> > "assumes that the underlying structure ... is known or can be accurately modeled. [which] can be challenging to define a priori in complex systems." [line: 354-357]
>
> We then propose "learning MAID structures directly from data" as a key direction for future work to overcome this precise limitation. We were encouraged that **Reviewer CWb2** found this discussion to be **"transparent and constructive,"** and we hope this clarification is helpful.
>
> ### **Re: The authors are encouraged to provide simple layman's explanations of their methodology.**
>
> Thank you for this suggestion. As advised, here is a simple, layman's explanation of our motivation, research question, and methodology, drawing from the example in our introduction.
>
> #### **Motivation and Research Question**
> As we state in our introduction, guiding a team of agents is often impractical when you can't give instructions to everyone.
>
> > Consider autonomous vehicles at a complex intersection. Providing specific feedback to all vehicles at once is infeasible due to safety validation and communication challenges. Yet, effective coordination is essential.
> >
> > This leads directly to our central research question: **What if providing an additional guidance signal to just one designated vehicle—perhaps to predictively adjust its speed—allowed other vehicles, by observing this targeted behavior, to adapt accordingly and foster smoother, safer passage for all?** Can we achieve effective team-wide coordination by assigning a targeted goal to just one agent?
>
>
>
> Our method, **Targeted Intervention**, provides a principled way to answer this question. It works in three steps:
>
> 1.  **Map the Team's Influence:** First, we use a graphical tool (a Multi-Agent Influence Diagram) to create a "flowchart" of the team's strategic dependencies. This map shows us which agent's decisions have the biggest ripple effect on the actions of others.
>
> 2.  **Give a Special Goal to One Agent:** Based on this map, we select a key agent and give it an additional goal on top of the main task. For the autonomous vehicle, this would be the "predictively adjust speed" instruction. For the card game Hanabi, it might be "always signal information about the most valuable cards."
>
> 3.  **Use a "Pre-Policy" to Guide the Agent:** We implement this with a learned component we call a **pre-policy module**. This module takes the special goal and translates it into a guidance signal that directly **influences the agent's decision-making process**, encouraging it to follow the instruction while still pursuing the main team objective.
>
>
> ### **Re: The core contribution is minimal, and results in only minor performance improvements.**
>
> We take the your concern about the significance of our contribution seriously. We **respectfully disagree with the assessment that the contribution is minimal**, and we would like to clarify the full scope of our work, which we believe is substantial.
>
> Our contribution is a complete, theory-to-practice framework:
> 1.  First, we **formalize** different multi-agent coordination paradigms using the principled graphical language of Multi-Agent Influence Diagrams (MAIDs).
> 2.  Second, based on this framework, we introduce our novel **Targeted Intervention** paradigm and a practical implementation via **pre-strategy intervention**, which is theoretically guaranteed to maximize the causal effect on a desirable system outcome.
> 3.  Third, we show how this theoretical concept can be **practically applied to MARL**, and our solvability analysis using relevance graphs provides a new way to predict the effectiveness of different learning approaches.
> 4.  Finally, we **verify** these theoretical claims with extensive experiments, showing that our method is not only effective but often superior to common alternatives.
>
> We are encouraged that other reviewers recognized the significance of this complete contribution. Specifically, Reviewer CWb2 highlighted that our method addresses the important challenge of "directing a complex system where universal intervention is unreasonable" and is "more scalable" while offering "both theoretical and practical analysis." We believe this consensus among other reviewers underscores the significance of our work.
>
>
>
> ### **Re: The Conclusion includes an ambiguous statement that gives the impression that the authors are the originators of the MAID framework.**
>
> We thank you for your careful reading, **but this could be a misunderstanding**. You are correct that MAIDs were invented by Koller and Milch (2003). **Our use of the word "introduced" was meant in the context of introducing MAIDs as a framework for our specific problem of  Targeted Intervention, not to claim their invention.**

---

> > ### Comment · Reviewer_djxC · 2025-08-01
> >
> > Thank you for the clarifications. I am having difficulty seeing how a Nash equilibrium is a better representation of external, human-defined goals than a pareto-optimal solution, since both are defined only in terms of the agents' individual reward signals and neither corresponds directly with human-perceived utility. It seems to me that the proposed framework will be most useful in contexts where a pareto-optimal solution is sought, as a Nash equilibrium is generally easy for independent and non-communicating agents to discover. By definition, a Nash equilibrium allows each agent to maximize its local reward independent of the actions of other agents; simple reinforcement learning methods can usually achieve this without the need for additional augmentations.
> >
> > Thank you for the reference to Section 6. I should more accurately have stated that the discussion of the paper's limitations is minimal. A more direct identification of real-world contexts in which MAID both can and cannot be expected to perform well would have been appreciated. A discussion of the implicit limitations raised by focusing on Nash vs. Pareto-optimal solutions would also have been valuable.

---

> > > ### Author Response · Authors · 2025-08-01
> > >
> > > We appreciate your prompt response and recognition for our rebuttal. However, we have some different idea about Nash equilibrium in multi-agent learning, as follows:
> > >
> > > We agree that your comments are correct in the traditional game theory, where information are sufficient to each player. Different from conventional settings of game theory where each player can know payoffs, during learning it should be noticed that **agents are not allowed to observe payoffs**. As a result, this hinders learning agents from easily converging to a Nash equilibrium. As far as we know, only special classes of games, for which learning agents can converge to a Nash equilibrium with special design of algorithms (e.g. potential game [1], two-player zero-sum game [2]). In a general-sum game, it is known that independent learning is not guaranteed to converge to a Nash equilibrium [3].
> > >
> > > **We believe the above discussion may validate that the framework proposed in our work is useful and non-trivial.** Ultimately, we are happy to have such an opportunity to exchange ideas with you.
> > >
> > > [1] Dong, J., Wang, B., & Yu, Y. (2024, April). Convergence to Nash Equilibrium and No-regret Guarantee in (Markov) Potential Games. In International Conference on Artificial Intelligence and Statistics (pp. 2044-2052). PMLR.
> > >
> > > [2] Perolat, J., Munos, R., Lespiau, J. B., Omidshafiei, S., Rowland, M., Ortega, P., ... & Tuyls, K. (2021, July). From poincaré recurrence to convergence in imperfect information games: Finding equilibrium via regularization. In International Conference on Machine Learning (pp. 8525-8535). PMLR.
> > >
> > > [3] Wunder, M., Littman, M. L., & Babes, M. (2010). Classes of multiagent q-learning dynamics with epsilon-greedy exploration. In Proceedings of the 27th International Conference on Machine Learning (ICML-10) (pp. 1167-1174).

---

> > > > ### Comment · Reviewer_djxC · 2025-08-01
> > > >
> > > > I see. Thank you for the clarification.

---

### Official Review · Reviewer_4sVj · 2025-07-01

**Clarity:** 3
**Significance:** 3
**Originality:** 3
**Rating:** 5
**Confidence:** 3

**Summary:**

The authors present MAIDs, a graphical tool to visualize existing coordination mechanisms among multi agents. Moreover, they design Target Intervention, a new coordination mechanism applied to a single agent. This algorithm is their answer to the challenge of achieving coordination through the stimulation of a single agent.
The paper formalizes MAIDs and the way MARL are represented using MAIDs. It also describes the pre-strategy intervention and demonstrates the causality of this representation.
Experiments are then conducted to prove the efficiency of the targeted intervention algorithm.

**Questions:**

In addition to the general comments, please find below some questions:
- In Fig 5, wouldn't it be expected to have MAPPO being outperformed by IPPO as MAPPO can be seen as a global intervention and IPPO as a direct one? Can you comment?
- In Section 5, wouldn't it be interesting to provide an analysis on the Nash Equilibrium (especially as it is detailed in the theoretical section)?

**Ethical Concerns:**

["NO or VERY MINOR ethics concerns only"]

**Final Justification:**

Authors have addressed most of my concerns. I maintain the score.

**Limitations:**

Yes

**Quality:**

3

**Strengths And Weaknesses:**

The paper is well written, and is very clear to read. The contributions are clearly listed, as well as the challenge the authors wish to tackle. Background and related work are very clearly described.
The principles of MAIDs and targeted intervention are extensively described, both in textual explanation and in mathematical forms.
The experimental setup is clearly described, making it doable for a reader to repeat the experiments. However, some of the methods mentioned in the baselines are not reported in the graphs.
In terms of results, the connection between the Z variable mentioned in the methodology and the results is not explicit. It would be worth elaborating on this aspect.
Conclusions and future work are very clear and include limitations.

---

> ### Author Rebuttal · Authors · 2025-07-30
>
> # Response to Reviewer 4sVj
> We sincerely thank you for your thorough review and positive assessment of our work.
>
> We are particularly encouraged that you found the paper to be "well written, and is very clear to read," and appreciated that our "contributions are clearly listed." We are also grateful for your recognition of the clear descriptions of our background, methodology, and experimental setup, which are crucial for reproducibility.
>
> This positive feedback is very valuable, and we also appreciate the insightful questions which we address below.
>
>
> ### **Re: In Fig 5, wouldn't it be expected to have MAPPO being outperformed by IPPO as MAPPO can be seen as a global intervention and IPPO as a direct one?**
>
> This is an excellent question that gets to the core of the practical challenges in MARL. The reviewer's intuition is correct: it is not always a given that a method with access to more "global" information like MAPPO will outperform a local information method like IPPO, especially in complex environments.
>
> While MAPPO's centralized critic is designed to mitigate the non-stationarity problem, this benefit comes at a significant cost in solving a game as complex as Hanabi: the **Curse of Dimensionality**.  The global observation spaces in Hanabi are enormous (over 1000 dimensions). MAPPO's centralized critic must learn a value function via exploring this massive joint space, which can be incredibly sample-inefficient and difficult to optimize. While a good solution may be guaranteed to exist in theory, the difficulty of searching this larger space means it is not guaranteed to be found in practice.
>
> At the same time, a naive direct interaction approach like IPPO also struggles. As our results in Figure 5 demonstrate, the baseline IPPO fails to converge to a good solution on its own, performing significantly worse than methods with guidance, such as the intrinsic reward approach or our own method.
>
> This highlights the core challenge that motivates our work: both complex centralization (MAPPO) and simple decentralization (IPPO) can fail to find a desirable equilibrium in challenging coordination games. This underscores the need for a more targeted and efficient guidance signal, as provided by our Targeted Intervention, which can be a more practical and effective approach to achieving coordination.
>
>
>
> ### **Re: In Section 5, wouldn't it be interesting to provide an analysis on the Nash Equilibrium.**
>
> We thank you for this excellent suggestion. We agree that a direct analysis of the learned equilibrium strengthens the connection between our theory and results. The Hanabi experiments provide a perfect case study for this.
>
> As established in the literature (e.g., "The Hanabi Challenge"[1]), the primary difficulty in Hanabi is its multiple, non-interchangeable equilibria. The challenge for MARL agents is to avoid converging to "inferior equilibria." For example, a team can get stuck in a stable but low-scoring "babbling equilibrium" where no agent gives meaningful hints because they expect to be ignored, and no agent listens because hints are uninformative. The goal is to find a high-performing, coordinated equilibrium instead.
>
> A specific human convention, such as "5 Save," represents exactly such a desirable, high-performing Nash Equilibrium [1, 2]. It is a shared strategy where no player has an incentive to unilaterally deviate, as doing so would break the team's coordination and lower the expected score.
>
> In our experiments, the intrinsic reward is explicitly designed to measure the agents' adherence to the target convention. Therefore, the intrinsic reward curves in our plots can be interpreted as a **direct, quantitative measure of the agents' convergence to this desired convention-based equilibrium.**
>
> As shown in Figure 5c (right plot), our Targeted Intervention method achieves a consistently high and stable intrinsic reward. This provides strong evidence that the agents have successfully learned and converged to the "5 Save" convention equilibrium. In contrast, the low and volatile intrinsic rewards for the baseline methods demonstrate their failure to converge to this desirable equilibrium, suggesting they fell into one of Hanabi's many inferior equilibria and were unable to establish a consistent, high-performing convention.
>
> [1] Bard N, Foerster J N, Chandar S, et al. The hanabi challenge: A new frontier for ai research[J]. Artificial Intelligence, 2020, 280: 103216.
>
> [2] H-Group Conventions https://hanabi.github.io/beginner
>
>
>
> ### **Re: Some of the methods mentioned in the baselines are not reported in the graphs.**
>
> We thank you for this comment and apologize for the lack of clarity. The plots in Figure 5 of the main paper were intended as a high-level summary due to space constraints, and we see now that this was unclear.
>
> To clarify, the curves labeled **"Global Intervention"** in Figure 5 directly represent the performance of the specific baseline methods we mentioned for that paradigm: **LIIR** and **LAIES**.
>
> For a complete view of our method's performance against every baseline algorithm mentioned in the text, please refer to the appendix, where we provide detailed plots for each individual comparison. For not reported  experiments in the main paper, the information can be found in the Appendix I, where we provide detailed plots comparing our method against each individual baseline algorithm mentioned in the text. Specifically:
>
>  * Figures 9 and 10 in the appendix show the per-algorithm results for MPE (vs. IQL, VDN, QMIX).
>
> * Figures 11 and 12 in the appendix show the per-algorithm results for Hanabi (vs. IPPO, MAPPO, PQN-IQL and PQN-VDN).
>
>
>
> ### **Re: The connection between the Z variable mentioned in the methodology and the results is not explicit.**
>
> Regarding the $Z$ variable, its practical implementation in our experiments is the intrinsic reward that guides the agent(s) toward the additional desired outcome. Specifically:
>
> * In our MPE experiments, $Z$ is the goal of reaching a specific landmark, and the intrinsic reward is the negative distance to it.
>
> * In our Hanabi experiments, $Z$ is the goal of adhering to a convention (like "5 Save"), and the intrinsic reward is a value indicating compliance with that convention.
>
> After $Z$ is determined, the measure of the goal (intirnsic reward) would be an input to pre-policy module at each step, as detailed in Appendix F.

---

> > ### Comment · Reviewer_4sVj · 2025-08-07
> >
> > Thank you for the answers.

---

### Official Review · Reviewer_CWb2 · 2025-07-01

**Clarity:** 3
**Significance:** 3
**Originality:** 3
**Rating:** 5
**Confidence:** 4

**Summary:**

The paper introduces a framework for guiding cooperative multi-agent reinforcement learning (MARL) systems using targeted pre-strategy intervention on a single agent within a coalition.  The authors use MAIDs to analyse the strategic dependencies within a multi-agent system and then based on this, they propose a “Targeted Intervention” paradigm which uses a pre-strategy intervention mechanism to maximise the causal effect on a desired system-wide outcome or goal. The key contributions of this paper are a theoretical analysis of different MARL paradigms using MAID-based relevance graphs and their solvability under different learning methods and a practical application of target pre-intervention in Hanabi and MPE which makes difficult problems more feasible for simple MARL algorithms.

**Questions:**

- The paper states that the experimental details and hyperparameter settings are in the appendix. For the sake of transparency and reproducibility, could you provide a more detailed breakdown of the results in the main paper, particularly for the MPE environment? Showing per-scenario performance would offer deeper insight into where the method excels or struggles, which is more informative than an aggregated curve.
- A central question left open by the "Targeted Intervention" framework is the selection criteria for the target agent. This seems especially critical in heterogeneous environments where agents may possess unique information or abilities, making the choice of whom to guide a non-trivial design decision. I would be interested in the authors' perspective on this:
  - How would you expect the method's performance to vary if a suboptimal agent were chosen as the target? An empirical analysis of this would greatly strengthen the paper.
  - Could the relevance graph be used to formalise a notion of "agent influence" to guide this selection?
  - Do you have any heuristics or preliminary principles for choosing the most effective agent to target?

**Ethical Concerns:**

["NO or VERY MINOR ethics concerns only"]

**Final Justification:**

The authors have provided new experimental results that address many of the concerns raised by the reviewers and have clearly explained how their method would be implemented in practice.

Although the authors do not introduce MAIDs,  it is important that we develop newer paradigms to train MARL agents, which, through future iteration, may lead to more powerful algorithms. I view the use of MAIDs as a retooling of existing methods to support the development of new types of algorithms by the authors for the paradigm of targeted-intervention, which does not seem as heavily explored as other MARL methods.

**Limitations:**

Yes, the authors have adequately addressed the limitations of their work in Section 6. They correctly identify that the framework currently assumes a known MAID structure and that their analysis focuses only on a single targeted intervention. They appropriately position these as key directions for future work, including learning the MAID structure from data and investigating the optimal criteria for selecting the target agent(s). The discussion is transparent and constructive.

**Paper Formatting Concerns:**

The line  “We introduced Multi-Agent Influence Diagrams (MAIDs) as a principled framework for Designing and analyzing Targeted Interventions in MARL systems.” could be misunderstood by readers as MAIDs being the primary contribution of the authors. This should be updated to be more clear in the final print.

The paper is also generally very dense and a simple diagram of the training method and architecture would greatly help a reader understand the practical implementation of algorithm along with a layman's explanation of the method to go along with the existing one.

**Quality:**

3

**Strengths And Weaknesses:**

## Strengths

- **Originality:** The paper introduces a novel framework that synthesises concepts for MAID, causal inference and MARL. This provides a principled justification for the new methods introduced by the authors and provides a new lens through which to view solving the challenge of guidance in MARL.
- **Significance:** Directing a complex system where universal intervention is unreasonable is addressed in this work. The method presented by the authors is more scalable and offers both theoretical and practical analysis of their contributions.
- **Quality and Clarity:** The paper is well written and easy to read. It also presents a well-founded theoretical framework to back its experiments. The use of MAID diagrams is also a clear and effective method to illustrate the coordination paradigms being studied.

## Weaknesses

- Agent selection criteria are undefined: The paper does not propose a method for selecting the target agent. This is limiting as the success of this method relies on effective agent selection.
- Lack of experiment details in the main paper. Key details for how the experiments are done have been deferred to the appendix and all reported performance is aggregate data. Presenting aggregate performance only may obscure performance nuances between different MPE scenarios.
- Experiments are only on homogeneous settings. Agent selection is less important in these settings.

---

> ### Author Rebuttal · Authors · 2025-07-30
>
> # Response to Reviewer CWb2
>
> We sincerely thank you for your thorough, insightful, and encouraging review. We are delighted that you found our framework to be novel, significant, and clear. Your constructive feedback has been invaluable, and we have conducted new experiments to directly address the weaknesses you identified.
>
> ### **Re: Showing per-scenario performance would offer deeper insight into where the method excels or struggles**
>
> We thank you for this excellent suggestion. You are correct that the main paper only presents the results for a single MPE scenario (Scenario 1) due to space constraints, and we agree that a per-scenario breakdown offers more valuable insight. As we cannot include images in the rebuttal, the table below summarizes the returns at each 10% of the training process.
>
>
> #### **Per-Scenario MPE Results (Extrinsic Returns)**
>
> | Timesteps | 10% | 20% | 30% | 40% | 50% | 60% | 70% | 80% | 90% | 100% |
> | :--- | :---: | :---: | :---: | :---: | :---: | :---: | :---: | :---: | :---: | :---: |
> | **Baseline** | -67.88 | -61.74 | -62.36 | -61.46 | -59.92 | -55.80 | -53.91 | -50.99 | -48.56 | -47.23 |
> | **Our Method (Scenario 1)** | -63.34 | -54.90 | -52.34 | -49.46 | -45.50 | -42.30 | -40.09 | -38.30 | -37.50 | -36.35 |
> | **Our Method (Scenario 2)** | -68.26 | -61.77 | -63.55 | -60.73 | -56.75 | -53.17 | -49.63 | -47.00 | -44.98 | -43.46 |
>
> #### **Per-Scenario MPE Results (Intrinsic Returns)**
>
> | Timesteps | 10% | 20% | 30% | 40% | 50% | 60% | 70% | 80% | 90% | 100% |
> | :--- | :---: | :---: | :---: | :---: | :---: | :---: | :---: | :---: | :---: | :---: |
> | **Baseline (Scenario 1)** | -24.47 | -22.00 | -24.41 | -23.56 | -24.56 | -23.67 | -23.46 | -23.35 | -22.99 | -22.66 |
> | **Our Method (Scenario 1)** | -15.68 | -11.21 | -10.32 | -10.14 | -10.17 | -9.96 | -9.84 | -9.44 | -9.50 | -9.39 |
> | **Baseline (Scenario 2)** | -25.72 | -21.47 | -22.38 | -21.64 | -20.59 | -19.26 | -17.85 | -16.94 | -15.92 | -15.40 |
> | **Our Method (Scenario 2)** | -24.15 | -20.98 | -21.64 | -20.43 | -18.50 | -16.92 | -15.55 | -14.57 | -13.94 | -13.46 |
>
>
> #### **Analysis**
> As a brief reminder of the two MPE scenarios: Scenario 1 provides the designated agent with an intrinsic reward for approaching a specific, predetermined landmark. In contrast, Scenario 2 provides an intrinsic reward for approaching the landmark that is farthest from its teammates.
>
> * In Scenario 1, we guide the target agent towards a fixed, predetermined landmark. The results show that this leads to a substantial and consistent improvement in the team's extrinsic return, significantly outperforming the baseline. This demonstrates that making one agent's behavior predictable can benefit the rest of the team to build a more effective coordinated strategy.
>
> * In Scenario 2, the additional goal is more complex, requiring the agent to reason about the target. While the absolute performance is lower than in the simpler Scenario 1, the results show that this "farthest landmark" strategy aligns well with an efficient team solution, even when the baseline agents begin to learn the solution emergently. Nevertheless, our intervention still provides a clear benefit by accelerating and refining the learning of this effective coordination strategy.
>
> These per-scenario results provide a clearer picture of our method's robustness, showing it can improve team performance. We also acknowledge that the more complex goal in Scenario 2 proved more challenging for the agents to learn, as reflected in the overall performance.
>
>
>
>
>
>
> ### **Re:  Selection Criteria for the Target Agent.**
>
> This is a central and very important question. We agree that providing an analysis of agent selection criteria, especially for heterogeneous environments, is crucial. Based on this feedback, we have developed a more formal perspective on agent influence and provide both a principled method and an intuitive case study below.
>
> #### **1. Formalizing "Agent Influence" using the Relevance Graph**
>
> As you suggest, the relevance graph is a very promising tool for formalizing a notion of "agent influence." A key agent can be identified by analyzing the graph's structural properties. While a simple heuristic is to measure an agent's direct influence via the out-degree of its decision nodes, a more concrete measure is its **total influence**, which considers all downstream decisions that are affected by its actions.
>
> Formally, let $\mathcal{D_i}$ be the set of decision variables for agent $i$, and let $Desc(D_k)$ be the set of all descendant nodes reachable from $D_k$ in the relevance graph. The total influence of agent $i$, denoted $I_{total}(i)$, can be quantified as the size of the union of the descendant sets of its decision nodes:
>
> $$I_{total}(i) = \left| \bigcup_{D_k \in \mathcal{D}_i} Desc(D_k) \right|$$
>
> This metric captures the entire chain of strategic dependencies that originate from agent $i$'s decisions. A higher $I_{total}(i)$ suggests that intervening on agent $i$ would have the broadest possible causal impact on the system.
>
> #### **2. Case Study: An Intuitive Example**
>
> To make our framework intuitive, imagine four friends planning a weekend trip:
>
> * **Alex (The Planner)** decides the **destination ($D_A$)**: "Beach" or "Mountains."
> * **Ben (The Driver)** decides the **vehicle ($D_B$)**: "SUV" or "Convertible."
> * **Chris (The Packer)** must choose the **shared gear ($D_C$)**, a decision that depends on both the destination ($D_A$) and the vehicle ($D_B$).
> * **Dana (The Booker)** must book **accommodation ($D_D$)**, which depends only on the destination ($D_A$).
>
>
> This planning process creates a relevance graph defined by the dependencies: $D_A \to D_C$, $D_A \to D_D$, and $D_B \to D_C$. Applying our influence metric:
>
> * **Alex's Influence (A):** The descendants of $D_A$ are $\{D_C, D_D\}$, so $I_{total}(A) = 2$.
> * **Ben's Influence (B):** The only descendant of $D_B$ is $D_C$, so $I_{total}(B) = 1$.
>
> This formal analysis reveals that intervening on **Alex (The Planner)** is the optimal choice, as their decision has the broadest impact on the team's plan. This case study demonstrates how our framework provides a principled, quantitative method for choosing between intervention targets.
>
> #### **3. Other principles and Future Work**
>
> In addition to this graph-theoretic approach, another formal principle we are exploring for future work is **empowerment** [1, 2], an information-theoretic measure of an agent's control, which may also correlate strongly with its ability to influence the system.
>
> [1] Klyubin A S, Polani D, Nehaniv C L. Empowerment: A universal agent-centric measure of control[C]//2005 ieee congress on evolutionary computation. IEEE, 2005, 1: 128-135.
>
> [2] Jaques N, Lazaridou A, Hughes E, et al. Social influence as intrinsic motivation for multi-agent deep reinforcement learning[C]//International conference on machine learning. PMLR, 2019: 3040-3049.
>
> ### **Re: Experiments are only on homogeneous settings.**
>
> We added new experiments in a heterogeneous MPE environment where agents have asymmetric speeds. Specifically, we created a scenario with one "sprinter" agent given 5x the normal speed, and two normal-speed agents. We then compared the team's performance when intervening on the sprinter versus intervening on one of the normal-speed agents.
>
>  **Heterogeneous MPE (Extrinsic Returns)**
>
> | Timesteps| 10% | 20% | 30% | 40% | 50% | 60% | 70% | 80% | 90% | 100% |
> | :--- | :---: | :---: | :---: | :---: | :---: | :---: | :---: | :---: | :---: | :---: |
> | **Baseline** | -68.74 | -63.19 | -66.18 | -62.42 | -57.25 | -53.24 | -49.92 | -46.89 | -44.24 | -41.87 |
> | **Intervene on Agent 0 (Sprinter)** | -61.35 | -59.70 | -59.03 | -54.49 | -49.00 | -46.20 | -40.34 | -37.72 | -36.18 | -35.05 |
> | **Intervene on Agent 1 (Normal)** | -65.71 | -62.44 | -60.86 | -55.85 | -50.43 | -44.79 | -40.87 | -36.59 | -35.16 | -34.05 |
>
>  **Heterogeneous MPE (Intrinsic Returns)**
>
> | Timesteps| 10% | 20% | 30% | 40% | 50% | 60% | 70% | 80% | 90% | 100% |
> | :--- | :---: | :---: | :---: | :---: | :---: | :---: | :---: | :---: | :---: | :---: |
> | **Baseline** | -41.90 | -25.28 | -67.59 | -47.68 | -39.75 | -29.30 | -27.79 | -27.10 | -27.20 | -23.94 |
> | **Intervene on Agent 0 (Sprinter)** | -18.06 | -18.23 | -17.48 | -14.88 | -13.06 | -12.64 | -10.91 | -10.24 | -9.77 | -9.51 |
> | **Intervene on Agent 1 (Normal)** | -20.83 | -19.47 | -19.28 | -17.71 | -15.52 | -13.27 | -12.29 | -11.46 | -11.30 | -10.96 |
>
> These results reveal a compelling insight: intervening on a normal-speed agent leads to superior team performance (extrinsic return) over intervening on the uniquely-fast "sprinter."
>
> While the sprinter easily achieved its individual goal (as shown by higher intrinsic rewards), the team's success was limited by the **coordination bottleneck** between the two slower agents. Our analysis shows that making one normal agent's behavior highly predictable allows the other to coordinate with it more effectively, directly addressing this primary bottleneck.
>
> This experiment provides strong evidence that our method works effectively in **heterogeneous teams**, and reveals that success comes from targeting the agent most critical to the team's coordination bottleneck, not necessarily the most individually capable one. While our current work demonstrates this principle, we agree that automatically identifying this key agent is an important direction for future work, for which information-theoretic metrics like **empowerment** are a promising approach.

---

### Official Review · Reviewer_rNPo · 2025-07-02

**Clarity:** 2
**Significance:** 2
**Originality:** 3
**Rating:** 4
**Confidence:** 4

**Summary:**

This paper proposes a novel method to improve zero-shot generalization in multi-agent reinforcement learning (MARL) via targeted intervention. The approach focuses on selectively modifying the observations or dynamics experienced by a specific agent during training, with the goal of improving coordination and generalization across a broader set of unseen scenarios. The method is motivated by the observation that intervening on certain "key agents" can have disproportionately positive effects on group performance. The paper provides theoretical motivation, proposes a framework for identifying candidate agents for intervention, and demonstrates empirical gains across multiple MARL environments.

**Questions:**

How is the targeted intervention actually implemented in the system? Please add an explicit algorithm or example to clarify.

How is the key agent identified? More formalization of this step would improve reproducibility and transparency.

What is the full training loop, including how and when interventions are applied? For example, are they applied during every episode? Is the policy conditioned on the presence of interventions?

What happens if the selected agent for intervention is trivial or uninformative?

**Ethical Concerns:**

["NO or VERY MINOR ethics concerns only"]

**Final Justification:**

Many of the issues I raised have been resolved through the authors' new experiments.

**Limitations:**

While the paper proposes an innovative framework, several limitations remain:

The effectiveness of the method may be highly sensitive to the choice of target agent for intervention. However, the paper does not provide an empirical analysis of what happens when a non-optimal or trivial agent is selected. This raises concerns about the method's robustness and reliability in more diverse or less structured environments.

The proposed approach is only tested on relatively small-scale MARL environments. It remains unclear whether the method can scale to systems with dozens or hundreds of agents, where the identification and intervention over a "key" agent becomes less straightforward.

While the core idea is intuitively compelling, the paper lacks theoretical analysis explaining under what conditions targeted interventions are effective.

Although the paper argues for improved generalization, it is not clear how broadly the proposed method can transfer to tasks with heterogeneous agent types, varying dynamics, or different coordination protocols.

The paper does not clearly explain how the intervention strategy itself is learned or tuned. If interventions are manually specified or heuristically chosen, this could limit the method’s adaptability and scalability to new domains.

**Paper Formatting Concerns:**

No.

**Quality:**

3

**Strengths And Weaknesses:**

The idea of targeted intervention is novel and addresses a major limitation of MARL: generalization to new agent configurations or dynamics. This could be impactful in settings such as swarm robotics or decentralized decision-making.

The paper offers compelling intuition that not all agents contribute equally to the system’s generalization ability, and that targeted manipulation of key agents can help bootstrap stronger coordination patterns.

Sections 3 and 4 are difficult to follow and feel disjointed. It is unclear how the proposed intervention mechanism is actually implemented and how it interfaces with the learning algorithm. A simple walkthrough or toy example would greatly help.

The learning pipeline remains vague. How is the intervention learned? Is it hardcoded, stochastic, or learned through a meta-objective? This lack of detail makes reproduction difficult.

While the idea is conceptually appealing, the theoretical justification for why and when targeted intervention leads to generalizable policies is weakly supported. The paper would benefit from more rigorous analysis or at least ablations across different agent heterogeneity levels.

---

> ### Author Rebuttal · Authors · 2025-07-30
>
> # Response to Reviewer rNPo
>
> Thank you for the detailed feedback and insightful questions. We are encouraged that you found our core idea "novel" and "impactful." The primary concerns regarding the clarity of our methodology, the need for a more thorough empirical analysis, and the strength of our theoretical justification are all valid points.
>
>
> ### **Re: How is the targeted intervention actually implemented in the system? Please add an explicit algorithm or example to clarify.**
>
> As stated in Line 305, the intervention is realized as a learned pre-policy module (an MLP/GRU). As detailed in Appendix H, this module takes the agent's observation and a measure of the additional goal as input, and its output embedding is used to influence the agent’s Q-value or critic function. For example, in the MPE environment, the pre-policy module inputs the targeted agent's observation and its distance to the selected landmark as the additional goal, and outputs an embedding to the agent's Q-value function. We prove in Appendix F that this practical implementation retains the same solvability properties as the original theoretical pre-strategy intervention.
>
>
> ### **Re: What is the full training loop, including how and when interventions are applied? ...  Is the policy conditioned on the presence of interventions?**
>
> **Our pre-policy module is learned end-to-end, not hardcoded.** The training loop operates as follows:
>
> At every timestep, our pre-policy module generates an embedding that is fed into the agent's critic or Q-value function. The entire system, both the agent policies and our intervention module, is then trained jointly using standard MARL algorithms to maximize a composite reward (Section 4.3). This reward combines the standard extrinsic (task) reward with an intrinsic reward derived from the additional goal.
>
> The additional goal itself is provided by a heuristic function, such as the distance to a landmark in MPE or the "Save 5" convention in Hanabi (detailed in Appendices H.2.1 and H.3.2).
>
> We agree that this reliance on human-defined heuristics is a limitation. However, as we stated in our future work section, we see great potential in using Large Language Models (LLMs) to automate the generation of these secondary goals. This would significantly enhance the autonomy of our framework.
>
>
> ### **Re: How is the key agent identified? More formalization of this step would improve reproducibility and transparency.**
> The key agent is identified based on its strategic influence, which we formalize using a **relevance graph** that maps the dependencies between strategies of agents. Our method requires selecting an agent that possesses outgoing edges in this graph, as this indicates its strategy directly influences the behavior of other agents, achieving coordination.
>
> In our current experiments, we select a fixed agent to intervene. This allows us to easily analyze the effectiveness of the intervention mechanism itself. This setup also covers many practical applications where a human operator would manually choose a specific agent to guide, confirming the applicability of our method in real-world systems.
>
>  ### **Re: What happens if the selected agent for intervention is trivial or uninformative?**
>
> This is an excellent question that directly addresses the importance of the "targeted" aspect of our method.
>
> In our framework, a "trivial" or "uninformative" agent is the one whose strategy has no direct influence on the rest of the team. This corresponds to a node in the **relevance graph** with no outgoing edges, as no other agent is **s-reachable** (Definition C.3) from it by definition.
>
> If such an agent was selected, the intervention would be ineffective. Any changes to its behavior would be isolated and would not propagate throughout the system to improve coordination.
>
>
> ### **Re: Sections 3 and 4 are difficult to follow and feel disjointed...A simple walkthrough or toy example would greatly help.**
> We are sorry for the lack of clarity. We have revised Sections 3 and 4 to improve the flow and better articulate the connection between our theory and its application.
>
> Our paper is structured to first establish a theoretical foundation and then show its practical implementation.
>
> * **Section 3** establishes a unified theoretical lens by showing how various MARL **coordination paradigms can be represented** using Multi-Agent Influence Diagrams (MAIDs). From this formal representation, we derive the corresponding relevance graph, which is the basis for our intervention mechanism.
>
> * **Section 4** then demonstrates how this framework extends to **sequential decision-making by proving the formal equivalence between MAIDs and Markov Games** (detailed in Appendix G), which justifies our method's application to MARL.
>
> In the revised manuscript, we have added a clearer narrative bridge to ensure this logical progression from theory to practice is easier to follow.
>
> ### **Re: It is not clear how broadly the proposed method can transfer to tasks with heterogeneous agent types**
>
> Thank you for the insightful question. To directly address the transferability of our method to heterogeneous agents, we added a new experiment in a modified MPE environment with asymmetric agent capabilities: one **"sprinter" agent** (5x speed) and two **normal-speed** agents. All methods, including our interventions and the baseline, are built upon the IQL algorithm. Other settings remain consistent with the experiments described in the main paper.
>
> We compared the team performance when intervening on the uniquely fast sprinter versus a normal-speed agent. The results are summarized below.
>
>
>  **Heterogeneous MPE (Extrinsic Returns)**
>
> | Timesteps| 10% | 20% | 30% | 40% | 50% | 60% | 70% | 80% | 90% | 100% |
> | :--- | :---: | :---: | :---: | :---: | :---: | :---: | :---: | :---: | :---: | :---: |
> | **Baseline** | -68.74 | -63.19 | -66.18 | -62.42 | -57.25 | -53.24 | -49.92 | -46.89 | -44.24 | -41.87 |
> | **Intervene on Sprinter** | -61.35 | -59.70 | -59.03 | -54.49 | -49.00 | -46.20 | -40.34 | -37.72 | -36.18 | -35.05 |
> | **Intervene on Normal Agent** | -65.71 | -62.44 | -60.86 | -55.85 | -50.43 | -44.79 | -40.87 | -36.59 | -35.16 | -34.05 |
>
>  **Heterogeneous MPE (Intrinsic Returns)**
>
> | Timesteps| 10% | 20% | 30% | 40% | 50% | 60% | 70% | 80% | 90% | 100% |
> | :--- | :---: | :---: | :---: | :---: | :---: | :---: | :---: | :---: | :---: | :---: |
> | **Baseline** | -41.90 | -25.28 | -67.59 | -47.68 | -39.75 | -29.30 | -27.79 | -27.10 | -27.20 | -23.94 |
> | **Intervene on Sprinter** | -18.06 | -18.23 | -17.48 | -14.88 | -13.06 | -12.64 | -10.91 | -10.24 | -9.77 | -9.51 |
> | **Intervene on Normal Agent** | -20.83 | -19.47 | -19.28 | -17.71 | -15.52 | -13.27 | -12.29 | -11.46 | -11.30 | -10.96 |
>
>
> These results reveal a compelling result: intervening on a normal-speed agent leads to superior team performance (extrinsic return) over intervening on the uniquely-fast "sprinter."
>
> While the sprinter easily achieved its individual goal (as shown by higher intrinsic rewards), the team's success was limited by the **coordination bottleneck** between the two slower agents. Our result shows that making one normal agent's behavior highly predictable allows the other to coordinate with it more effectively, directly addressing this primary bottleneck.
>
> This experiment provides strong evidence that our method works effectively in **heterogeneous teams**, and reveals that success is attributed to targeting the agent that is most critical to the team's coordination bottleneck, rather than the most individually capable one.
>
>
>
>
> ### **Re: The proposed approach is only tested on relatively small-scale MARL environments.**
>
> To address the valid concern about scalability, we have also added new experiments on the 4-player version of Hanabi based on PQN-VDN. All other experimental settings and hyperparameters were held constant. We believe this is a significant extension, because as highlighted by recent work, the complexity of Hanabi's coordination problem increases dramatically with the number of players[1].
> #### **4-player version Hanabi Performance Results (Extrinsic Returns)**
>
> | Timesteps      | 10%      | 20%      | 30%      | 40%      | 50%      | 60%      | 70%      | 80%      | 90%      | 100%     |
> | :----------- | :------: | :------: | :------: | :------: | :------: | :------: | :------: | :------: | :------: | :------: |
> | Baseline MARL | 8.58     | 11.19    | 11.96    | 11.90    | 15.33    | 14.01    | 14.90    | 15.51    | 15.98    | 17.70    |
> | Global Intervention| 8.03     | 10.73    | 11.49    | 13.67    | 13.68    | 14.27    | 16.57    | 15.48    | 15.81    | 17.64    |
> | Our Method | 12.08| 14.70| 17.29| 18.26| 18.96| 19.44| 19.78| 20.24| 20.43| 20.18|
>
> Our method's strong performance in the 4-player setting, achieving a final score of **20.18** compared to the baseline's **17.70**, provides a strong evidence to show its ability to scale and handle more complex coordination challenges. We will include these results in our revised manuscript.
>
> [1] Sudhakar A V, Nekoei H, Reymond M, et al. A Generalist Hanabi Agent[C]//ICLR. 2025.
>
>
> ### **Re: The theoretical justification for why and when targeted intervention leads to generalizable policies is weakly supported.**
> Our method is most effective when there are strong strategic dependencies, a condition we analyze using **relevance graphs (Section 4.2)**. These dependencies manifest as cycles in the graph that impede independent learners. Our intervention **breaks these cycles** by making a targeted agent's behavior predictable, transforming the problem into a more solvable, acyclic one.
>
> This works because the intervention is explicitly optimized to maximize its **causal effect** on the team's outcome—a property we formally guarantee in Proposition 3.5. We verify this empirically by showing our method allows a decentralized learner (IQL) to match the performance of a centralized one (VDN).

---

> > ### Comment · Reviewer_rNPo · 2025-08-05
> >
> > I appreciate the authors' efforts in addressing the concerns. After reviewing their responses, I have decided to raise my score.

---

> > > ### Author Response · Authors · 2025-08-05
> > >
> > > Thank you for reconsidering our work and for raising your score. We're grateful for your insightful feedback and are confident that the new experiments and clarifications will significantly strengthen the final paper.

---

### Decision · Program_Chairs · 2025-09-17

**Decision:**

Accept (poster)

**Comment:**

This paper presents a method using targeted interventions on single key agent in a MARL setting to improve performance. The approach builds on the Multi-Agent Influence Diagram (MAID) framework, formalizing MARL in this context along with describing the approach, and demonstrating performance.

There are some concerns about the presentation and the novelty of using MAIDs but the author response and discussion was helpful in terms of presentation questions and while MAIDs isn't a contribution, its use in MARL to build the proposed intervention method is. The paper should be updated to clarify some of the assumptions about the intervention (e.g., selecting the target agent and heuristics), solution concept, and implementation details.